# COMP-1 promotes competitive advantage of nematode sperm

**Jody M Hansen, Daniela R Chavez, Gillian M Stanfield\***

Department of Human Genetics, University of Utah, Salt Lake City, United States

**Abstract** Competition among sperm to fertilize oocytes is a ubiquitous feature of sexual reproduction as well as a profoundly important aspect of sexual selection. However, little is known about the cellular mechanisms sperm use to gain competitive advantage or how these mechanisms are regulated genetically. In this study, we utilize a forward genetic screen in *Caenorhabditis elegans* to identify a gene, *comp-1*, whose function is specifically required in competitive contexts. We show that *comp-1* functions in sperm to modulate their migration through and localization within the reproductive tract, thereby promoting their access to oocytes. Contrary to previously described models, *comp-1* mutant sperm show no defects in size or velocity, thereby defining a novel pathway for preferential usage. Our results indicate not only that sperm functional traits can influence the outcome of sperm competition, but also that these traits can be modulated in a context-dependent manner depending on the presence of competing sperm.

## Introduction

Sexual selection operates at the level of reproductive success to promote traits that improve offspring production (*Darwin, 1871*). It thus influences a wide array of processes that affect not only the likelihood of mating, but also the probability that gametes will interact within a female to form a viable zygote. In many species, a female can mate with multiple males, resulting in competition between male ejaculates, known as sperm competition (*Parker, 1970*). In addition, having multiple mates provides opportunities for a female to influence the outcome, known as cryptic female choice (*Eberhard, 1996*). These post-copulatory forms of sexual selection have driven the diversification of sperm and reproductive tract morphologies as well as the divergence of reproductive proteins, and have likely contributed to reproductive isolation and speciation (*Ritchie, 2007*; *Howard et al., 2008*; *Manier et al., 2013*).

Sperm competition is a widespread phenomenon that occurs in species utilizing a wide range of reproductive strategies, and a variety of different patterns of preferential usage, generally referred to as precedence, have been observed. (*Smith, 1984*; *Birkhead and Møller, 1992*, *1998*). For example, in some species, the first male to mate may show precedence, while in others, the last mate's sperm may win, and the strength of such effects varies widely. By their nature, events in the reproductive tract that determine the outcome of competition are difficult to study, so in most cases the mechanistic basis for a particular precedence pattern is poorly understood. When sperm competition is intense, males often respond by production and transfer of numerous smaller sperm (*Gomendio et al., 1998*; *Simmons, 2001*). However, in some cases, sperm may gain an advantage by modulating functional traits, for example, by increasing migration velocity, promoting retention, or blocking subsequent access to the site of fertilization (*Wigby and Chapman, 2004*; *Gomendio and Roldan, 2008*; *Pizzari and Parker, 2009*).

Due to the difficulty of distinguishing sperm from different ejaculates or of observing sperm directly within the selective environment, indirect assays have often been employed to measure sperm usage. The cell behaviors underlying sperm competition have only been investigated in a few species

**\*For correspondence:** gillians@
genetics.utah.edu

**Competing interests:** The authors declare that no competing interests exist.

**eLife digest** The ornate feathers of a peacock and the antlers of a stag are both traits that have evolved because they help a male to outcompete his rivals and mate with more females. Similarly, in species where each female mates with multiple males, a male can improve his reproductive success if his sperm outcompetes the other males' sperm and fertilizes the female's eggs. One way that males try to gain a competitive edge is by producing large quantities of sperm. However, it is also possible that males could compete by generating higher-quality sperm, for example, cells that are better at migrating through the reproductive tract.

A microscopic worm called *Caenorhabditis elegans* is often used to investigate sperm competition. Each worm is either a hermaphrodite or a male; and hermaphrodites store their sperm within a storage structure and use it later to fertilize their own eggs. However, if a hermaphrodite mates with a male, the male's sperm displaces the hermaphrodite's stored sperm and fertilizes the eggs instead. The male's sperm cells were thought to be more competitive because they are larger and faster than the hermaphrodite's sperm, but recent findings suggest a more complex scenario.

Hansen et al. identified a genetic mutation that causes male *C. elegans* sperm cells to lose their competitive advantage. Male worms with a mutation in a gene called *comp-1* produce sperm cells that are normal in size, but that cannot outcompete sperm from a non-mutant hermaphrodite. Although sperm cells from a *comp-1* mutant male can migrate through the reproductive tract and fertilize eggs when other sperm are not present, in competitive situations the mutant sperm cells have difficulties migrating and are often absent from the hermaphrodite's sperm storage structure. Thus, they no longer come into contact with the eggs that they seek to fertilize.

Sperm cells from hermaphrodite *comp-1* mutants have similar defects; but when a *comp-1* mutant male mates with a *comp-1* mutant hermaphrodite, the mutant male sperm regains its fertilization advantage. The identification of the *comp-1* gene provides a preview into a complex network of environmental cues and genetically encoded traits that influence which sperm cells are most likely to fertilize an egg cell, and thus live on in the next generation.

that are amenable to such analyses, and little is known about the genetic basis for differences in competitive ability among cells. However, in vivo imaging studies have recently begun to reveal the cellular mechanisms of sperm behavior in competitive contexts, where multiple males have mated with a female (e.g., *Civetta, 1999*; *Manier et al., 2010*; *Marie-Orleach et al., 2014*). For example, in *Drosophila*, analyses of genetically labeled fluorescent sperm have revealed that stored sperm are highly motile and that modulation of sperm storage, release, and ejection by the female contribute strongly to second-male precedence in that organism (*Manier et al., 2010*; *Lupold et al., 2012*). Some genetic loci that affect male reproductive success have recently been identified in *Drosophila* and in mammals (e.g., *Fiumera et al., 2005*; *Sutton et al., 2008*; *Yeh et al., 2012*; *Civetta and Finn, 2014*). Specific seminal fluid components have been shown to play an important role in male competitive advantage by affecting sperm motility and storage, as well as female responses (e.g., *Mueller et al., 2008*; reviewed in *Avila et al., 2011*; *Simmons and Fitzpatrick, 2012*). However, very few examples are known of genes that function in sperm to control characteristics directly involved in sperm competition. An open question is whether genes exist that specifically regulate competition, without affecting core sperm functions, or whether competitive advantage is always gained by modulating the activity of genes involved in other processes.

The nematode *Caenorhabditis elegans* provides a model system to address the cellular behaviors and molecular pathways that mediate sperm competition. *C. elegans* is a male-hermaphrodite species in which hermaphrodites produce their own self sperm but also can be inseminated by males. In a self-fertilizing context, hermaphrodite self sperm reside in the spermathecae, sperm storage organs where fertilization occur, and are used with very high efficiency. Typically, more than 99% of sperm go on to fertilize an oocyte (*Ward and Carrel, 1979*). However, if mating occurs, male sperm migrate through the uterus to the spermathecae, where they encounter and must compete with stored self sperm. Importantly, during male-hermaphrodite sperm competition, male sperm are used preferentially (*Ward and Carrel, 1979*; *LaMunyon and Ward, 1995*). Male precedence is very robust, and many crosses result in male sperm exclusively fertilizing oocytes. Simple numerical advantage, seminal fluid

factors, and the order of introduction into the reproductive tract have been ruled out as potential causes (*Ward and Carrel, 1979*; *LaMunyon and Ward, 1994*, *1995*). Instead, the competitive advantage of *C. elegans* male sperm has been shown to rely on intrinsic differences between male and hermaphrodite sperm cells. While the form of male and hermaphrodite sperm is the same, male sperm are generally larger than hermaphrodite sperm (*LaMunyon and Ward, 1999*). Consistent with the idea that this is significant, experimental evolution under crossing conditions has been shown to lead to increased sperm size (*LaMunyon and Ward, 2002*). Like those of other nematodes, *C. elegans* sperm move by crawling using a pseudopod, and this motility is required for precedence (*Nelson et al., 1982*; *Singson et al., 1999*). Larger sperm crawl faster in vitro (*LaMunyon and Ward, 1998*), and male sperm displace self sperm from the walls of the spermathecae (*Ward and Carrel, 1979*). However, male sperm need not fertilize oocytes to outcompete hermaphrodite sperm; mutant males whose sperm are motile, but fertilization-defective, block self progeny production even though their sperm cannot be used (*Singson et al., 1999*). These data suggest a model for male precedence in which the presence of larger, faster, male sperm leads to the exclusion of self sperm from the fertilization process (*LaMunyon and Ward, 1998*; reviewed in *Ellis and Stanfield, 2014*). Differences in the migration behaviors of male and hermaphrodite sperm could affect the processes of sperm migration towards, retention in, or localization within the spermathecae, where there could be sites especially favorable for sperm–egg interaction (*Han et al., 2009*). Although many mutants defective for spermatogenesis and/or fertilization have been identified in genetic screens, most mutations affect both male and hermaphrodite sperm equally and none specifically affect male precedence (reviewed in *Nishimura and L'Hernault, 2010*). Thus, the underlying mechanisms, in terms of either cellular behaviors or genetic controls, remain unclear.

In this study, we report the use of a genetic screen in *C. elegans* to identify a sperm competition gene. While sperm lacking *comp-1* activity are used efficiently in the absence of competition, *comp-1* sperm are outcompeted by wild-type sperm from either hermaphrodites or males, resulting in reduced reproductive success for both *comp-1* mutant males and the hermaphrodites that mate with them. Strikingly, *comp-1* sperm are normal in size. However, they show defects in sperm motility and storage in vivo, coupled with context-dependent defects in pseudopodial extension in vitro. These results suggest a model in which *comp-1* functions in sperm to coordinate environmental signals that influence motility-related functions required for sperm to compete with one another. Our findings provide key insight into the genetic regulation of sperm competition and suggest that in *C. elegans*, sperm gain advantage by modulating their motility and storage depending on their competitive milieu.

## Results

### Isolation of a *C. elegans* mutant with defects in male precedence

We took advantage of the male-hermaphrodite reproductive system and robust sperm precedence order of *C. elegans* to perform a forward genetic screen for males with less-competitive sperm. After a wild-type male mates with and transfers sperm to a hermaphrodite, his sperm rapidly migrates to the spermathecae and begins fertilizing oocytes, and in ideal conditions, most crosses result in more than 90% cross progeny (*Ward and Carrel, 1979*; *LaMunyon and Ward, 1995*). However, the underlying mating and sperm transfer behaviors are variable in efficiency, so that in practice a wide range of cross-progeny frequencies are often observed, and some crosses fail altogether (*Ward and Carrel, 1979* and unpublished observations). Thus, for our screen, we developed a sperm competition assay, using *spe-8; dpy-4* hermaphrodite recipients, that allowed us to exclude crosses for which cross progeny numbers were decreased due to behavioral defects (*Figure 1A*, 'Materials and methods'). *spe-8* hermaphrodites are self-sterile due to a defect in the ability to activate their self sperm to become motile (*L'Hernault et al., 1988*). In the absence of mating, they produce no offspring. However, if a male mates with and transfers seminal fluid to a *spe-8* recipient, both the male and self sperm are activated to become motile and fertilization-competent, and since male sperm are superior, they fertilize the vast majority of oocytes (*LaMunyon and Ward, 1995*). The *dpy-4* mutation is recessive and allows discrimination of self progeny from cross progeny on the basis of the Dumpy phenotype. For our assay, we established mating conditions in which most crosses were successful and fewer than five total Dumpy self progeny were produced in the vast majority of cases, providing a readily scored cutoff for candidate mutants (data not shown).

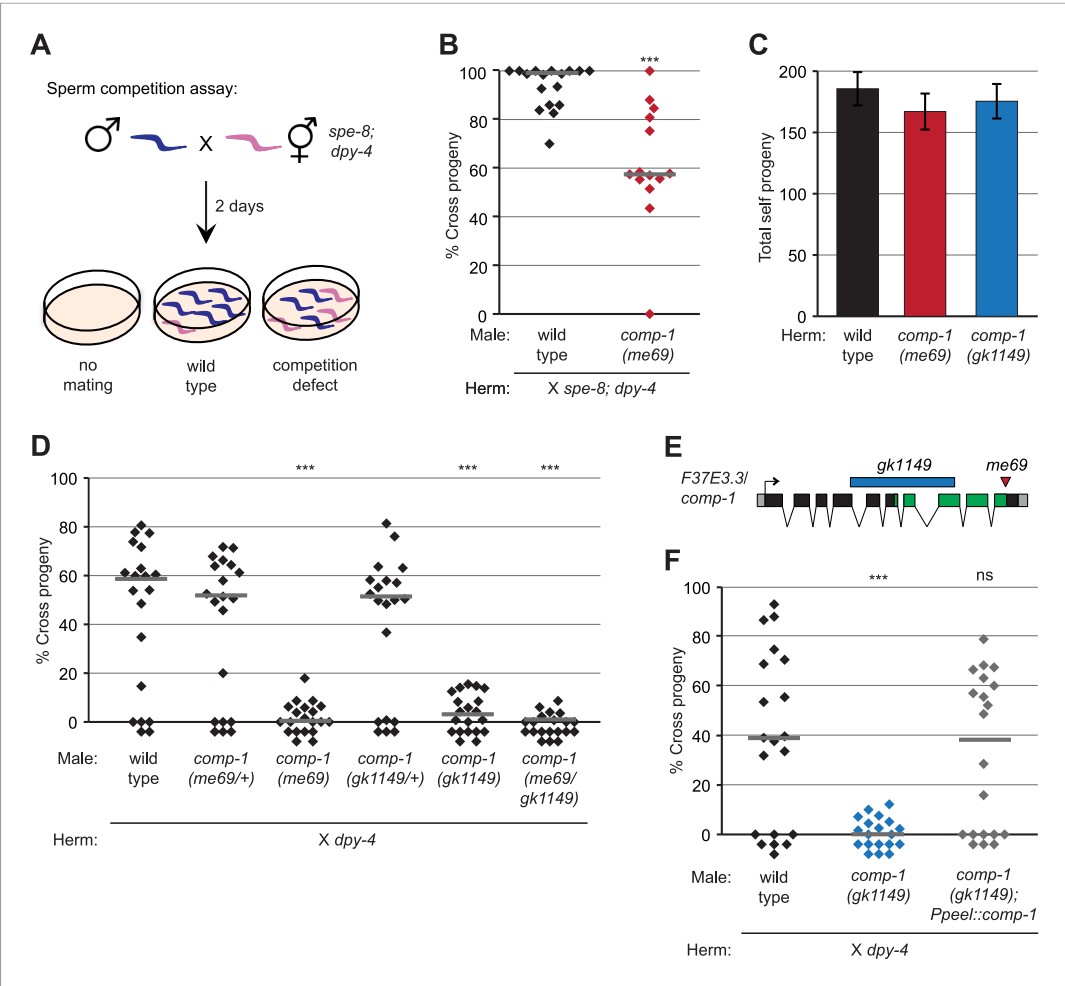

**Figure 1**. Isolation of the male precedence mutant *me69* in a genetic screen. (**A**) Screening assay for mutants with reduced male precedence, showing outcomes for mating failure, mating by wild-type males, and mating by males with less-competitive sperm. (**B**) *me69* males have decreased precedence in the screen assay. Males were mated to *spe-8(hc53); dpy-4* hermaphrodites, and offspring were scored as Dumpy (self) or non-Dumpy (cross) progeny. (**C**) *me69* and *gk1149* mutant hermaphrodites have normal self fertility. Total self progeny of unmated hermaphrodites were counted. Error bars, 95% confidence intervals; p > 0.05 (Student's t test). (**D**) Mutants for *comp-1* have defects in male precedence. Males were mated to *dpy-4* hermaphrodites, and offspring were scored as Dumpy (self) or non-Dumpy (cross). (**E**) Schematic of the *F37E3.3* gene showing the kinase-like domain (green) predicted by the Conserved Domains Database (*Marchler-Bauer et al., 2011*) and the locations of the *me69* and *gk1149* alleles. Boxes, exons; black, coding regions; gray, 5′ and 3′UTRs. (**F**) Expression of the *F37E3.3* gene in sperm rescues the *comp-1* male precedence defect. Male precedence was assayed for *comp-1(gk1149); jnSi168[Ppeel::comp-1]* and control strains as in *Figure 1D*. (**B**, **D**, **F**) Each point represents the result of an individual cross; lines indicate medians. ***, p < 0.001; **, p < 0.01; ns, not significant (Kolmogorov–Smirnov test; all comparisons are to wild-type). In addition to the genotypes shown, all males were homozygous (**B**, **C**, **F**) or heterozygous (**D**) for *him-5 (ok1896)*, and control strains in (**F**) harbored *oxSi221*.

The following figure supplements are available for figure 1:

**Figure supplement 1**. COMP-1 is highly conserved within the *Caenorhabditis* genus and present in related parasitic species.

**Figure supplement 2**. COMP-1 transgenes rescue the male precedence defects of *comp-1* mutants.

We performed EMS mutagenesis on a male-producing *him-5* strain (*Hodgkin et al., 1979*), established lines from individual F2 hermaphrodites, and tested F3 males from each line in the sperm competition assay. We identified one mutant, *me69*, which showed reduced male precedence as compared to the wild type (*Figure 1B*; GMS, unpublished data). The percentage of cross progeny that resulted from mating with *me69* mutant males was rarely comparable to that of wild-type crosses. However, *me69* mutant hermaphrodites produced a normal number of offspring (*Figure 1C*), setting the *me69* phenotype apart from those of previously identified *spe* mutants, most of which were isolated based on the reduction of hermaphrodite fertility but usually affect sperm production in males as well (*Nishimura and L'Hernault, 2010*).

While the use of *spe-8* recipients was critical for our screen, their immotile self spermatids cannot maintain proper positioning within the reproductive tract, resulting in mislocalization and gradual loss of the sperm to the external environment (*L'Hernault et al., 1988*). To assess precedence of *me69* males in a more natural competitive context, we performed crosses to *dpy-4* hermaphrodites, whose sperm localize appropriately to the spermathecae (data not shown). We placed individual L4 males and *dpy-4* hermaphrodites together for 40 hr and quantified self and cross progeny generated during this time period. Under these conditions, most matings with wild-type control males resulted in at least some cross progeny, and most successful males sired a high fraction of offspring (*Figure 1D*). However, matings with *me69* males resulted in few or no cross progeny during the time frame of this assay. We confirmed that *me69* males were capable of mating and transferring sperm to these hermaphrodites at a high frequency (48–85% of crosses were successful, as compared to 74–100% for wild-type), so their poor reproductive success was not simply due to behavioral defects. Rather, *me69* mutant males show post-copulatory defects in sperm usage consistent with a defect in male precedence.

## The *me69* mutation disrupts *comp-1*, a kinase domain gene that functions in sperm

We used meiotic mapping to localize *me69* to a 6.7-Mb interval on chromosome I (*Davis et al., 2005*, *Tables 1 and 2*). Whole-genome sequencing of the *me69* strain revealed a likely candidate for the causal mutation as a G to A transition in the coding region of *F37E3.3*, an uncharacterized gene that we have renamed *comp-1* for sperm *comp*etition defective (*Figure 1E*). Based on global expression analyses, *comp-1* is expressed in the germ line during time periods that coincide with sperm production: the L4 larval stage in hermaphrodites and in both L4 and adult males (*WormBase*; *Reinke et al., 2000*, *2004*; *Ortiz et al., 2014*).

The COMP-1 protein contains divergent SH2 and protein kinase-like domains and has been classified within a 'unique' subset of *C. elegans* kinases that do not fall clearly within defined families (*Manning, 2005*); it also lacks closely related paralogs within the *C. elegans* genome. It is missing three highly conserved core motifs present in active kinases, including the VAIK motif in the N lobe, the HRD motif in the catalytic loop, and the DFG motif within the activation loop, though it does contain the tripeptide APE motif located within the activation segment (*Figure 1—figure supplement 1*) (*Hanks et al., 1988*; *Hanks and Hunter, 1995*; *Manning et al., 2002*; *Nolen et al., 2004*; *Marchler-Bauer et al., 2011*). The absence of these features suggests that the protein is unlikely to have catalytic activity. The *me69* allele is predicted to result in a glycine to arginine change in a residue that is conserved in all other orthologs identified to date. COMP-1 orthologs are present in other *Caenorhabditis* species as well as in the parasites *Haemonchus contortus*, *Ancylostoma ceylanicum*, and *Necator americanus* (*Figure 1—figure supplement 1*) (*WormBase*; *Laing et al., 2013*; *Schwarz et al., 2013*; *Tang et al., 2014*). Although COMP-1 appears to be absent from more distant species (*WormBase*, and unpublished data), it is present in nematodes that utilize male-female as well as male-hermaphrodite reproductive modes.

We obtained a *comp-1* deletion allele, *gk1149*, from the *C. elegans* Deletion Mutant Consortium (*C. elegans Deletion Mutant Consortium, 2012*). *gk1149* eliminates a large region of the coding sequence and is likely a null allele. To test if the *me69* and *gk1149* alleles result in a similar male precedence defect, we crossed *gk1149* males to *dpy-4* hermaphrodites and found that *gk1149* mutant males indeed showed a reduction in male precedence as compared to the wild type (*Figure 1D*). Like *me69*, *gk1149* is recessive; crosses with heterozygous *gk1149/+* males showed a wild-type precedence pattern. However, *me69/gk1149* heterozygotes had male precedence defects, indicating the two mutations failed to complement one another. To confirm that loss of

*comp-1* function is responsible for the male precedence defect, we performed rescue experiments. We generated animals harboring a Mos-mediated single copy insertion (MosSCI) transgene (*Frøkjær-Jensen et al., 2008*, *2012*) encompassing a 3.9-kb genomic fragment surrounding *F37E3.3* (*Tables 3 and 4*). This transgene rescued the male precedence defect of both *me69* and *gk1149* males (*Figure 1—figure supplement 2*), confirming that *comp-1* is the gene affected in these mutants.

To test if *comp-1* function is required in sperm cells, we generated MosSCI transgenes to express it specifically in sperm, using the promoter for the *peel-1* gene (*Seidel et al., 2011*). We observed full rescue of the male precedence defect in *comp-1(gk1149); Ppeel-1::comp-1* males (*Figure 1F*), indicating that expression of *comp-1* in sperm is indeed sufficient to rescue the male precedence defect. Thus, *comp-1* acts in sperm to promote their preferential usage.

## *comp-1* activity influences the outcome of male–male sperm competition

Since COMP-1 is highly conserved in both male-hermaphrodite and male-female species (*Figure 1—figure supplement 1*), we hypothesized that *comp-1* might function in male–male sperm competition. In the standard laboratory strain of *C. elegans* (N2), sequential male matings normally show no precedence pattern, i.e., the first and second males to transfer sperm are equally likely to sire offspring (*Ward and Carrel, 1979*; *LaMunyon and Ward, 1998*). However, sequential matings of males from different wild-type strains can show preferential sperm usage patterns (*LaMunyon and Ward, 1998*; *Murray et al., 2011*), indicating that differences in competitive ability can occur among males in this species. To determine if *comp-1* function influences sperm competition in a male vs male context, we performed sequential matings of wild-type and/or *comp-1* males to *fog-2* mutant hermaphrodites, which fail to produce self sperm and are essentially female (*Schedl and Kimble, 1988*). To facilitate assignment of paternity, we used strains containing a GFP transgene, *mIs11*, for either the first or second sets of crosses, and scored offspring for the presence or absence of fluorescence. In control crosses, in which two wild-type males were sequentially mated to hermaphrodites, progeny numbers from the first and second male were variable, but no consistent bias was observed, other than a weak trend in which non-*mIs11* males seemed to be slightly favored over *mIs11*-containing males (*Figure 2A*, *Figure 2—figure supplement 1*). Similarly, in sequential matings of two *comp-1* males, no precedence order was observed. However, sequential matings of wild-type and *comp-1* males resulted in strong precedence for the wild-type sperm, regardless of whether wild-type males were the first or second mates. Notably, *comp-1* males showed full fertility in crosses to *fog-2* hermaphrodites, which lack their own sperm (*Figure 2B*). These data indicate that *comp-1* males transfer normal numbers of functional sperm, which can be used efficiently when they do not need to compete. However, when other sperm are present, *comp-1* sperm show poor usage. Furthermore, the reduced usage of *comp-1* sperm is unrelated to the order of their introduction into the hermaphrodite reproductive tract. Rather, male sperm lacking *comp-1* function appear to have an intrinsic disadvantage as compared to wild-type sperm.

## *comp-1* male sperm are not used until hermaphrodite self sperm are depleted

To investigate the importance of *comp-1* activity for male reproductive success, we sought to determine the nature of the competitive defect of *comp-1* mutant sperm. In particular, we wished to know if *comp-1* sperm were lost or remained active within the gonads of hermaphrodites. To address this question, we assayed the long-term kinetics of usage of *comp-1* male sperm within hermaphrodites. We crossed wild-type or *comp-1* males to *dpy-4* hermaphrodites for 16 hr, transferred the recipients at 12-hr intervals until they ceased egg laying, and counted the total number of self and cross progeny at each time point. Wild-type male sperm usage increased rapidly after mating (*Figure 3A*), consistent with previous evidence that male sperm are used preferentially over hermaphrodite self sperm (*Ward, 1977*; *Ward and Carrel, 1979*; *LaMunyon and Ward, 1995*). However, *comp-1* mutant males sired almost no progeny until late in the hermaphrodite lifespan (*Figure 3A,B*). Furthermore, while mating with wild-type males suppressed usage of self sperm, mating with *comp-1* males had no effect on self-progeny production (*Figure 3C*). Thus, *comp-1* males show severe long-term defects in their ability to produce offspring after mating, and hermaphrodites that mate with *comp-1* males produce a decreased number of total offspring (*Figure 3D*). However, although they are initially unsuccessful in fertilizing eggs, at least some *comp-1* sperm are eventually used, indicating that they can remain in the reproductive tract.

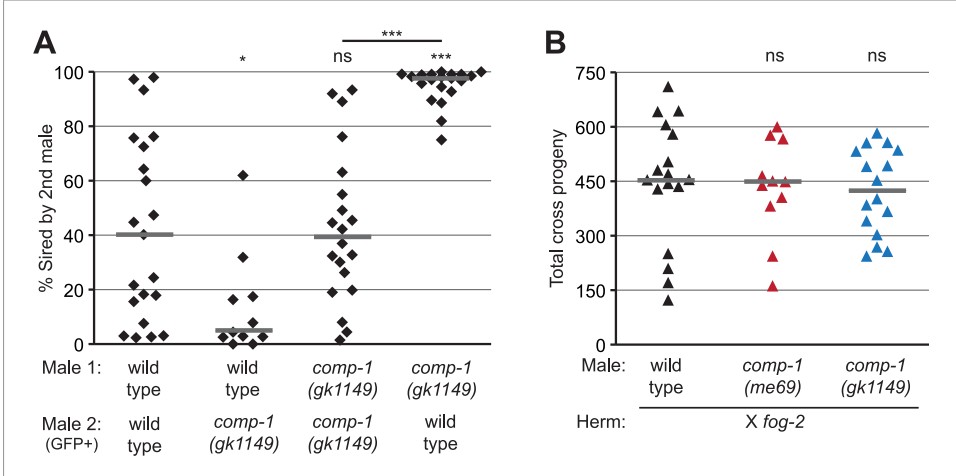

**Figure 2**. The *comp-1* mutant has defects in male–male sperm competition. (**A**) *comp-1* male sperm are outcompeted by wild-type male sperm. Wild-type and/or *comp-1(gk1149)* males were mated sequentially to *fog-2* hermaphrodites; second-mated males harbored the transgene *mIs11*(GFP+). Offspring were scored for GFP, and the percentage of GFP-positive progeny produced 0–16 hr after second-male mating is shown. (**B**) *comp-1* mutant males have wild-type levels of fertility in the absence of competition. Males were crossed to *fog-2* hermaphrodites and total progeny were counted. (**A**, **B**) Individual data points are shown; lines indicate medians. *, p < 0.05; ***, p < 0.001; ns, not significant (Kolmogorov–Smirnov test; comparisons are to wild-type unless indicated by a line linking the two data sets).

The following figure supplement is available for figure 2:

**Figure supplement 1**. The *comp-1* mutant has defects in male–male sperm competition.

Since hermaphrodites make their entire store of self sperm prior to oocyte production, they gradually run out of self sperm during adulthood. The onset of *comp-1* sperm usage correlated with the depletion of stored hermaphrodite self sperm, suggesting that although *comp-1* sperm remained present from matings that occurred at an earlier time, they were only used once fewer self sperm were present to compete with (*Figure 3B,C*; compare time points at 52 and 76 hr). To test if *comp-1* sperm can be used more rapidly when fewer self sperm are present, we aged hermaphrodites until they had used up part or all of their self sperm reservoir, then crossed them to males and assessed the short-term usage of male sperm. In crosses to 12, 24, and 36 hr post-L4 recipients, which retain moderate levels of self sperm (see *Figure 3E*; 'No. remaining sperm'), the number of offspring sired by *comp-1* males increased proportionally to the age of the hermaphrodite, but success was always reduced as compared to wild-type males (*Figure 3E*). However, in crosses performed with 48 hr post-L4 hermaphrodites, which have nearly run out of self sperm, *comp-1* males produced as many offspring as the wild type. Thus, regardless of the length of time they have been resident in the reproductive tract, *comp-1* sperm are unsuccessful specifically in situations where other sperm are present, but can be used in the absence of competition.

### *comp-1* is expressed and functions to promote sperm usage in both sexes

Global expression studies suggested that *comp-1* is expressed not only in males, but also in hermaphrodites (*Reinke et al., 2000, 2004*). We sought to determine if COMP-1 is indeed present in hermaphrodite sperm and if it shows any differences in localization as compared to male sperm. We first generated transgenic animals carrying a *Pcomp-1::GFP::H2B* transcriptional reporter, in which GFP localizes to the nuclei of *comp-1*-expressing cells. Expression was visible in developing spermatocytes and spermatids in both males and hermaphrodites, and we observed no obvious differences in abundance between the two sexes (*Figure 4A–D* and data not shown).

To determine the localization of COMP-1, we generated worm strains expressing transgenes that contained the full-length *comp-1* coding region fused to either mCherry or GFP. Worms carrying the

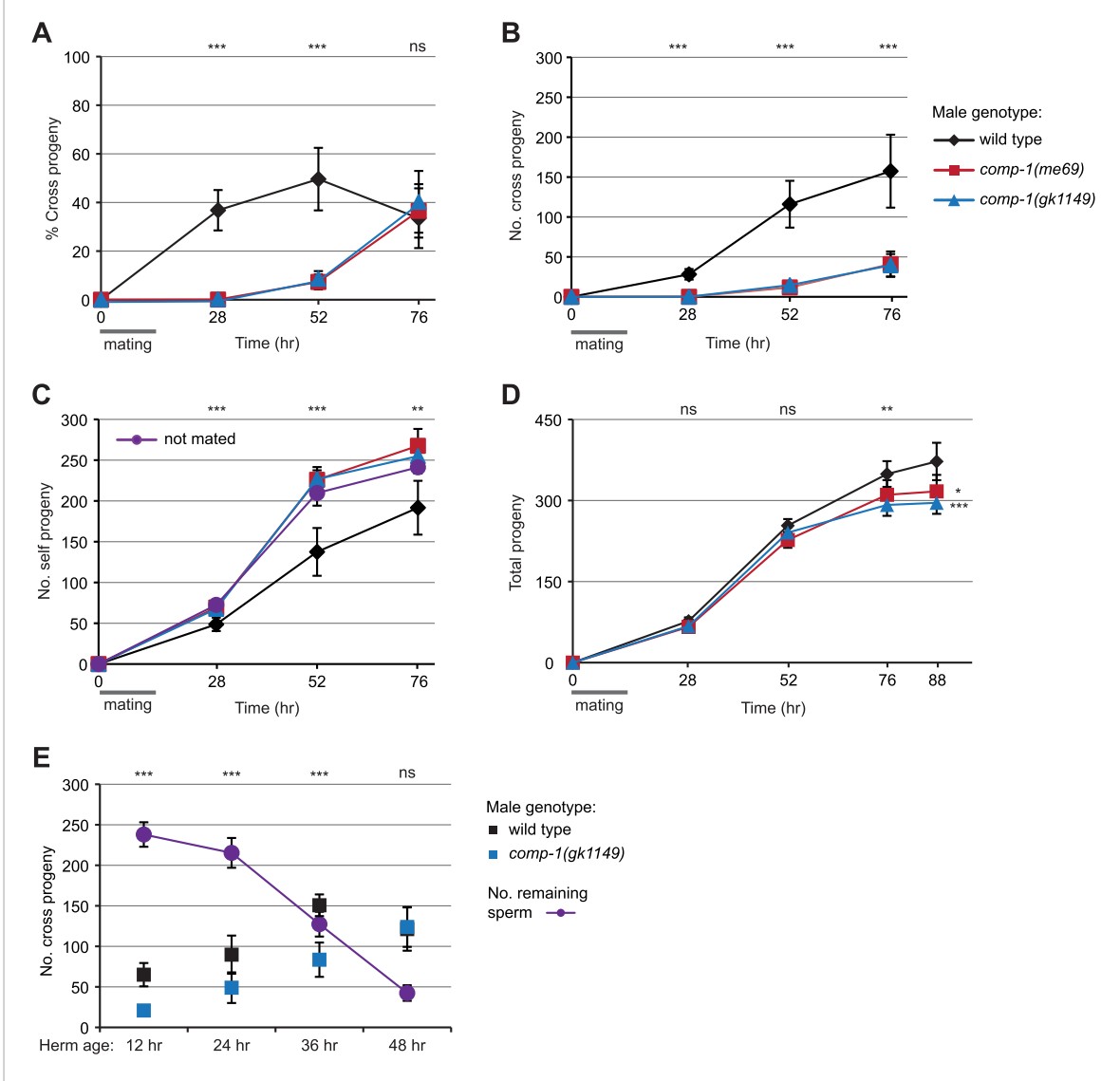

Figure 3. *comp-1* male sperm have long-term precedence defects. (**A**) Crosses with *comp-1* males result in a low percentage of cross progeny. (**B**) The number of cross progeny sired by *comp-1* increases at late time points. (**C**) Crosses with *comp-1* males do not suppress production of self progeny. Purple line indicates self progeny of unmated hermaphrodites. (**D**) Crosses with *comp-1* males result in decreased progeny numbers as compared to those with wild-type males. (**A–D**) Males were crossed to *dpy-4* hermaphrodites for 16 hr (gray line); progeny were collected throughout the recipients' reproductive lifespans and scored as self or cross progeny. All graphs are from a single data set that is representative of three repeats. For **B–D**, cumulative progeny numbers are shown. (**E**) *comp-1* male sperm are used at wild-type levels in crosses to sperm-depleted hermaphrodites. Males were crossed to staged *dpy-4* recipients for 24 hr and progeny generated during the mating period were scored as self or cross progeny. 'No. remaining sperm' indicates the number of self sperm present within recipients at each stage, inferred from brood counts of unmated *dpy-4* hermaphrodites performed in parallel. Data points indicate averages; error bars, 95% confidence intervals. **, p < 0.01; ***, p < 0.001; ns, not significant (Kolmogorov–Smirnov test, comparing wild-type to each *comp-1* mutant).

GFP fusion showed rescue of the male precedence defect, suggesting that the fluorescent tags did not interfere with protein function or localization (**Figure 4—figure supplement 1** and data not shown). The COMP-1 fusion proteins displayed a punctate pattern in the cytoplasm of both developing spermatids and mature sperm, where they were restricted to the cell body region (**Figure 4E–F** and data not shown). These punctae were visible in sperm from both males and hermaphrodites (**Figure 4** and data not shown). To determine whether the COMP-1 protein was localized to a specific subcellular location, we performed co-labeling experiments with the vital dye

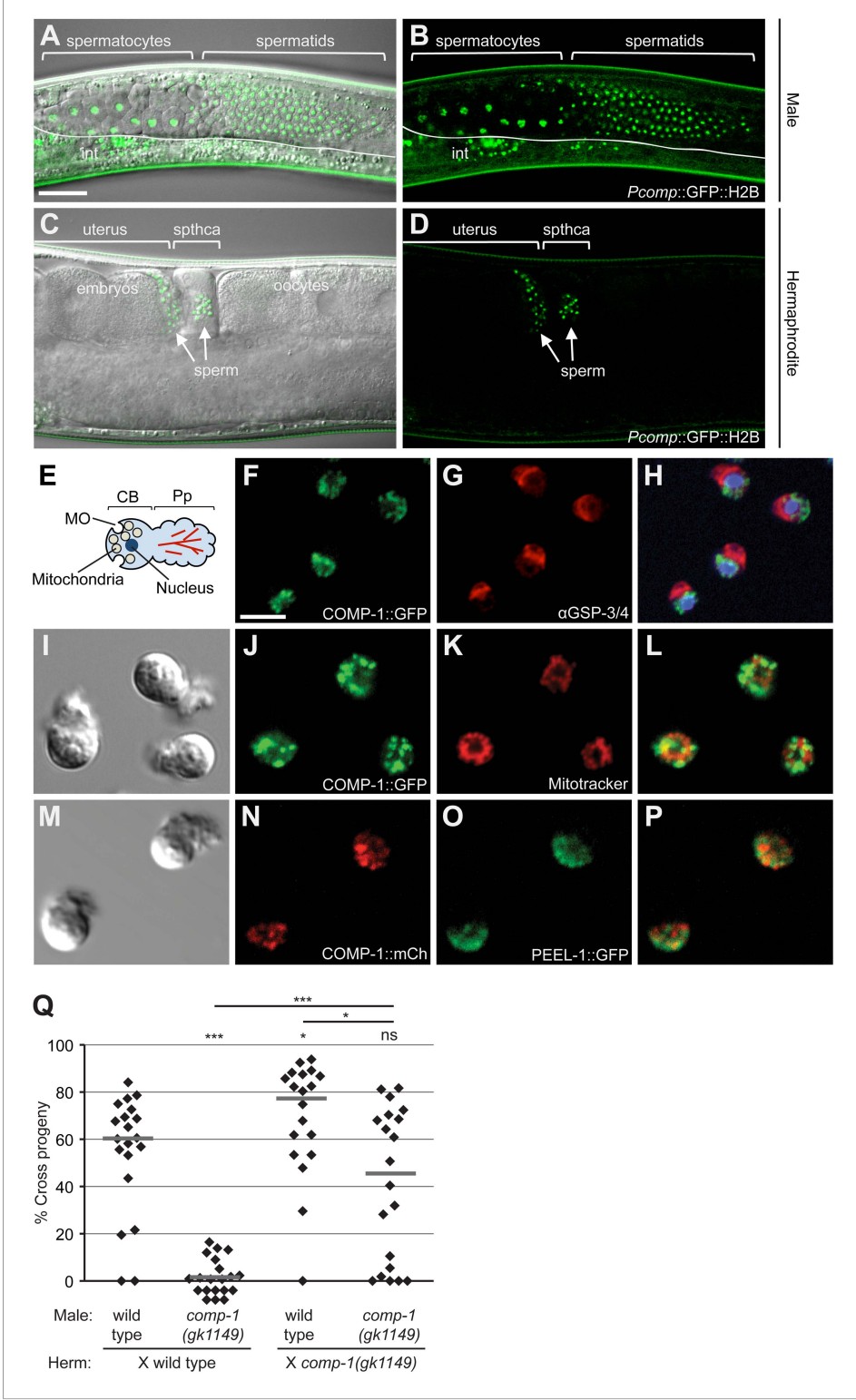

**Figure 4**. COMP-1 is expressed and functions in sperm of both males and hermaphrodites. (**A–D**) Images of *jnSi118 [Pcomp-1::GFP::H2B]* adult males (**A**, **B**) and hermaphrodites (**C**, **D**), which express the *comp-1* reporter in developing sperm. int, intestinal autofluorescence. Scale bar (**A–D**), 30 μm. (**E**) Schematic of structural organization of spermatozoa. (**F–H**) COMP-1 does not colocalize with GSP-3/4, which is in the pseudopod. Images of *jnSi171 [COMP-1::GFP]* male spermatozoa fixed and stained with α-GSP-3/4 antibody (red) and DAPI (blue). Scale bar (**F–P**), 5 μm. (**I–L**) COMP-1 does not colocalize with mitochondria. Images of *jnSi171[COMP-1::GFP]* male spermatozoa
*Figure 4. continued on next page*

*Figure 4. Continued*

stained with Mitotracker. (**M–P**) COMP-1 does not colocalize with PEEL-1::GFP, which is at the membranous organelles. Images of *jnSi143[COMP-1::mCherry]; jnSi177[PEEL-1::GFP]* male spermatozoa. (**Q**) *comp-1* functions in both male and hermaphrodite sperm. Wild-type and *comp-1(gk1149)* males were tested against wild-type and *comp-1(gk1149)* hermaphrodites in the short-term precedence assay. Lines indicate medians. *, p < 0.05; ***, p < 0.001; ns, not significant, Kolmogorov–Smirnov test.

The following figure supplement is available for figure 4:

**Figure supplement 1**. A COMP-1::GFP transgene rescues the male precedence defects of *comp-1* mutants.

Mitotracker, a marker of mitochondria, and PEEL-1::GFP, which labels the sperm-specific membranous organelles (MOs) (*Chen et al., 2000*; *Seidel et al., 2011*). We also examined the phosphatase GSP-3/4, which is involved in cytoskeletal dynamics and shows polarized localization within the pseudopod (*Wu et al., 2011*). COMP-1 did not colocalize with any of these markers of sperm structure or with the sperm nucleus (*Figure 4E–P*), and its absence from the pseudopod suggests that it is not involved directly with cellular locomotion, at least by modulating cytoskeletal dynamics.

The absence of obvious differences between males and hermaphrodites in the expression and localization of COMP-1 raised the question of a potential role for *comp-1* in hermaphrodite self sperm. We thus assayed precedence of wild-type and *comp-1* mutant males in crosses to *comp-1* hermaphrodites. Matings of wild-type males to *comp-1* hermaphrodites resulted in even higher levels of cross progeny production than those seen in crosses to wild-type hermaphrodites, consistent with *comp-1* hermaphrodite sperm having reduced ability to compete (*Figure 4Q*). Interestingly, when *comp-1* males were mated to *comp-1* hermaphrodites, mutant male sperm usage was indistinguishable from that of wild-type, suggesting that the male precedence order is regained when *comp-1* sperm compete against each other. These results indicate that *comp-1* functions to promote sperm usage not only in males, but also in hermaphrodites. In addition, factors other than *comp-1* must influence the outcome of competition, as a strong precedence effect can be observed in the absence of its activity in both competing populations of sperm.

## *comp-1* function is not required for sperm development

One potential explanation for the male precedence defect in *comp-1* mutants was that sperm do not undergo proper spermatogenesis or spermiogenesis necessary to mature into functional sperm. Loss of function of *spe* or *fer* genes required for spermatogenesis generally leads to hermaphrodite self sterility and male infertility, and reduction of gene function can result in partial fertility (*Kadandale and Singson, 2004*; *Nishimura and L'Hernault, 2010*). We thus examined available markers of sperm morphology to determine if *comp-1* sperm harbor any general defects. Males and hermaphrodites both produce immotile, spherical spermatids that must be activated to become mature, pseudopod-bearing sperm competent for motility and fertility (*Wolf et al., 1978*). *comp-1* mutant spermatids and sperm appear grossly normal by light microscopy (*Figure 5A,C*; *Figure 5E,G* and data not shown). In addition, several markers of sperm structures localized appropriately in the *comp-1* mutant. As in wild-type sperm, mitochondria and membranous organelles were restricted to cell bodies (*Figure 5A–H*), and the GSP-3/4 phosphatase was polarized within pseudopodia (*Figure 5I–L*). The presence of properly polarized sperm structures in mutant sperm indicates that *comp-1* is not required to complete spermatogenesis, nor is it necessary for proper localization of sperm structures in mature sperm cells. These findings are consistent with the absence of fertility or sperm usage defects in *comp-1* animals in the absence of competition.

## *comp-1* promotes competitive ability independently of cell size

In *C. elegans*, a key factor in conferring male precedence is thought to be the differential size of male and hermaphrodite sperm cells. Male sperm are generally larger than hermaphrodite sperm, correlating with faster crawling speeds in vitro, and growth under conditions with a high risk of sperm competition has been shown to result in increased sperm size (*LaMunyon and Ward, 1998*, *2002*).

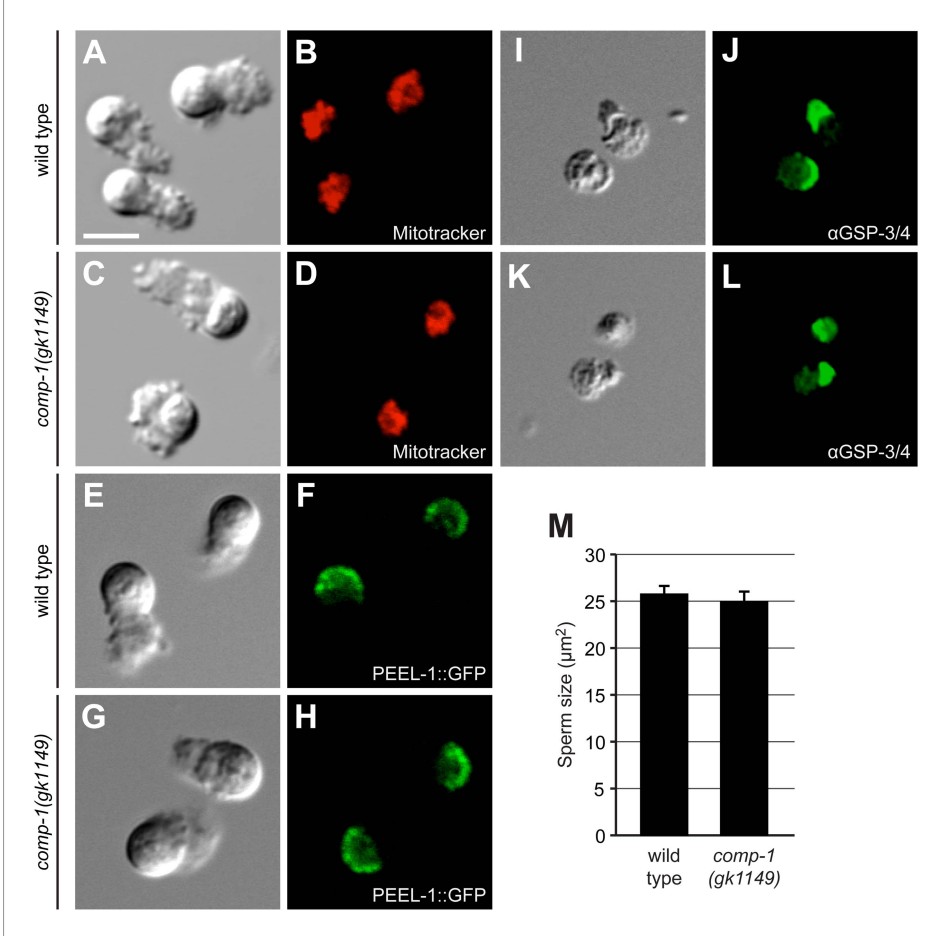

**Figure 5**. *comp-1* sperm have normal organization and size. (**A–D**) Wild-type (**A**, **B**) and *comp-1(gk1149)* (**C**, **D**) spermatozoa stained for mitochondria using Mitotracker. (**E–H**) Wild-type (**E**, **F**) and *comp-1* (**G**, **H**) spermatozoa expressing PEEL-1::GFP (membranous organelles). (**I–L**) Wild-type (**I**, **J**) and *comp-1(gk1149)* (**K**, **L**) spermatozoa fixed and stained with α-GSP-3/4 antibody (green). (**A–L**) Scale bar, 5 µm. (**M**) *comp-1* male spermatid size is not significantly different from wild-type. Cross sectional areas through the center of spermatids were measured. Error bars, 95% confidence interval; p = 0.41, Student's t test.

Thus, we investigated the possibility that the precedence defects of *comp-1* males might be due to a reduction in the size of mutant sperm. To assay cell size, we measured spermatids, which are spherical, by obtaining a cross-sectional area through the center of each cell (*LaMunyon and Ward, 1998*). *comp-1* mutant spermatids were variable in size, but the average and distribution of their sizes were indistinguishable from those of wild-type spermatids (*Figure 5M*). Therefore, we conclude that loss of *comp-1* does not reduce competitive ability by affecting cell size. Furthermore, *C. elegans* sperm can achieve precedence by a size-independent mechanism.

## *comp-1* is required for efficient migration to and localization within the spermathecae

In *C. elegans*, sperm are stored and fertilization occurs within the spermathecae (*Ward and Carrel, 1979*). Transferred male sperm must migrate through the uterus and into the spermathecae to be eligible to fertilize oocytes, and male sperm have been observed to displace hermaphrodite sperm from the walls of these structures (*Ward and Carrel, 1979*). We thus examined the ability of *comp-1* mutant sperm to migrate toward and access the spermathecae. We crossed unlabeled hermaphrodites to males either labeled with Mitotracker dye (*Kubagawa et al., 2006*; *Stanfield and Villeneuve, 2006*) or expressing a sperm *H2B::GFP* reporter, and then examined male sperm positioning at

different time points after transfer to the hermaphrodite reproductive tract. Similar to previously reported analyses of sperm migration (*Kubagawa et al., 2006*), we divided each proximal gonad arm into four regions: zone 1, near the sperm entry point at the vulva; zone 2, within the uterus; zone 3, the region near the spermatheca; and the spermatheca itself (*Figure 6A*).

By 1–1.5 hr after transfer, a majority of wild-type male sperm had migrated to zone 3 and the spermatheca (*Figure 6B*). Some crosses with *comp-1* males also showed this pattern. However, in many cases, a large percentage of *comp-1* sperm remained in zone 1 and/or zone 2, and accumulation in zone 3 and the spermatheca was reduced (*Figure 6C*). Importantly, wild-type and *comp-1* sperm were present in similar, high numbers (an average of $222.4 \pm 78.6$ for wild-type, n = 13; an average of $191.1 \pm 63.3$ for *comp-1*, n = 14), and there was no obvious correlation between the number of sperm transferred and their migration efficiency (data not shown). Sperm-specific expression of *comp-1*

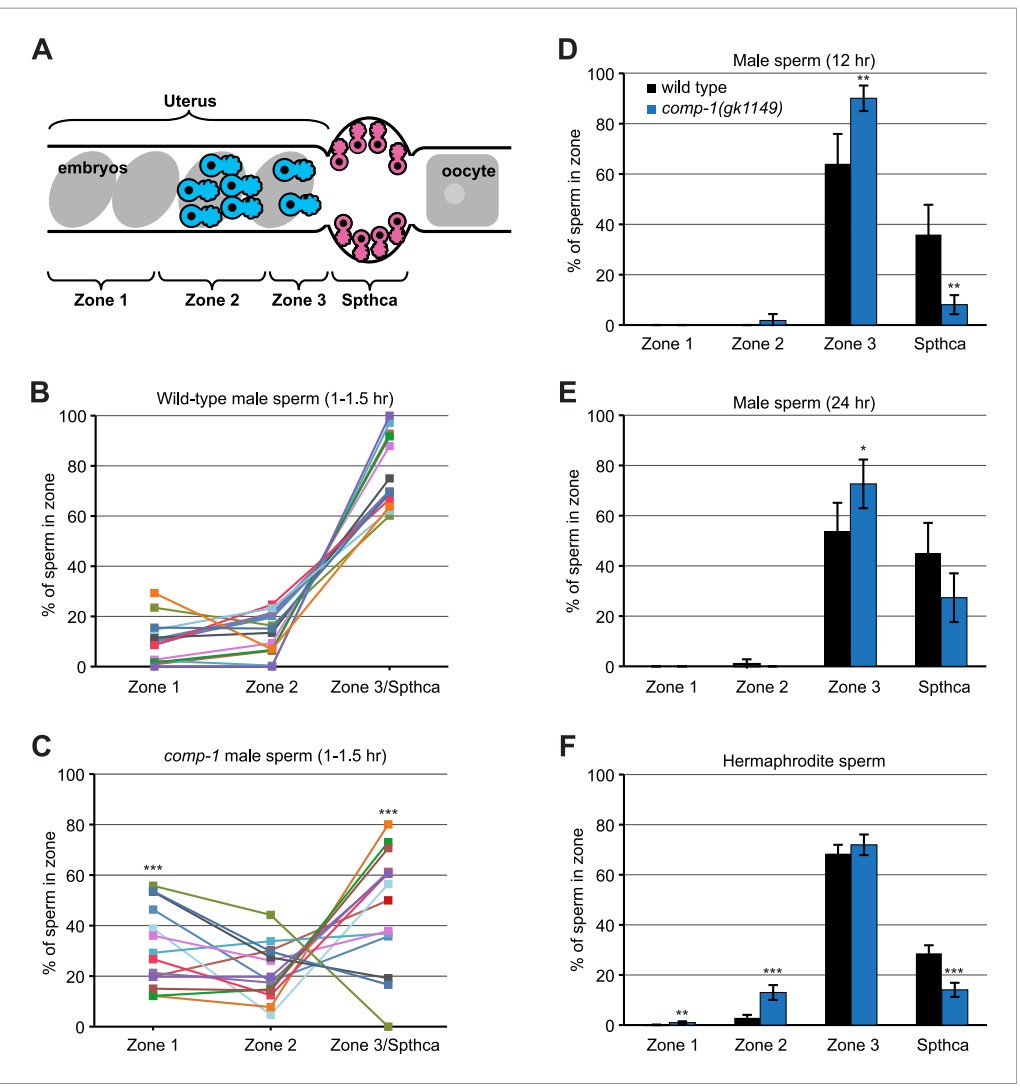

**Figure 6.** *comp-1* sperm have defects in migration and spermathecal accumulation. (**A**) Schematic of hermaphrodite gonad arm showing zones used to quantify sperm position. (**B**, **C**) Localization of wild-type (**B**) and *comp-1(gk1149)* (**C**) Mitotracker-labeled male sperm 1–1.5 hr after transfer to hermaphrodites. Percentage of total male sperm is shown. (**D**, **E**) Localization of *jnSi118[GFP::H2B]* male sperm 12 hr (**D**) and 24 hr (**E**) after transfer to hermaphrodites. Percentage of male sperm in the focal plane with maximum sperm in the spermatheca is shown. (**F**) Localization of hermaphrodite self sperm in 24 hr post-L4 unmated hermaphrodites. Animals were stained with DAPI to facilitate counting of sperm cells. Percentage of total hermaphrodite sperm is shown. Error bars, 95% confidence intervals. *, p < 0.05; **, p < 0.01; ***, p < 0.001; Student's t test.

rescued the migration defect, confirming that the altered migration was due to loss of *comp-1* (data not shown). By 12 hr after transfer, both wild-type and mutant sperm were rarely found in zones 1 and 2, instead localizing to zone 3 and/or the spermatheca (*Figure 6D*; see below). Thus, mutant male sperm show a delay in reaching the spermathecal region. However, their ability to accumulate near the spermathecae at later time points indicates that they are competent to respond to directional cues.

In addition to this delay in migration, we observed a significant decrease in residency of *comp-1* sperm within the spermathecae. At 12 hr after transfer, when wild-type sperm consistently occupied the spermathecae, very few *comp-1* sperm localized there, even though they were present in zone 3 (*Figure 6D*). Interestingly, by 24 hr post-mating, mutant male sperm numbers increased within the spermathecae and there was little, if any, difference between wild-type and *comp-1* sperm positions (*Figure 6E*). This later time point corresponded to 48 hr post-L4 adult hermaphrodites, in which self sperm numbers are largely depleted and *comp-1* male sperm start to show increased usage (*Figure 3B,E*). Taken together, these results suggest that mutant male sperm are not used because they are present at lower numbers in the spermathecae during periods when these structures are occupied with large numbers of self sperm. Since fertilization can occur only within these structures, this defect is likely the primary reason for the reduction in the competitive ability of *comp-1* sperm.

Since *comp-1* functions in both male and hermaphrodite sperm, we also analyzed self sperm in *comp-1* hermaphrodites to assess whether localization defects might still be present in a non-competitive context. We quantified the position of sperm in different zones in DAPI-stained 24 hr adult hermaphrodites. In wild-type hermaphrodites, most of the sperm resided in zone 3, tightly concentrated just outside of the spermathecae (*Figure 6F* and data not shown); a smaller number was present within the spermathecae. In *comp-1* hermaphrodites, while the majority of sperm were localized within zone 3, fewer sperm resided within the spermathecae as compared to wild-type. In addition, some *comp-1* self sperm were mislocalized to zone 2 and occasionally zone 1. Thus, *comp-1* hermaphrodite sperm have minor defects in localization and spermathecal residency that are similar to those of *comp-1* male sperm. However, these defects do not result in reduced fertility.

## *comp-1* sperm have context-dependent defects in pseudopodial extension

To probe the cellular basis for the localization defects of *comp-1* sperm, we analyzed their motility using established in vivo and in vitro assays (*Geldziler et al., 2011*). Measured immediately after transfer, the migration velocities of *comp-1* sperm within the hermaphrodite uterus were indistinguishable from those of wild-type (*Figure 7A*). Furthermore, migrating *comp-1* sperm showed highly directional movement through the uterus towards the spermathecae (*Figure 7A*) and a low reversal frequency, consistent with guided migration (among cells analyzed for motility, only 3/28 wild-type cells and 1/25 *comp-1* cells showed one or more reversals during the assay period). The ability of *comp-1* sperm to migrate rapidly in vivo suggests that basal motility is not affected in the mutant. However, the difference between wild-type and *comp-1* mutant sperm migration patterns could be due to aspects of other migratory behaviors, such as the amount of time individual sperm spend actively migrating through the reproductive tract.

To further analyze the motility of *comp-1* mutant sperm, we dissected spermatids, treated them with the known in vitro activators TEA (triethanolamine, a weak base) or Pronase (a protease mixture) (*Ward et al., 1983*; *Shakes and Ward, 1989*), and sought to measure the velocities of cells crawling on glass slides (*Nelson et al., 1982*). *comp-1* sperm activated in TEA had extended pseudopods and were capable of crawling at speeds similar to those of wild-type cells (*Figure 7B–D*, *Figure 7—figure supplement 1A*). *comp-1* sperm treated with Pronase activated at rates similar to the wild type (*Figure 7—figure supplement 1B*), based on the presence of a pseudopod in the majority of cells. However, the shapes of *comp-1* cells were markedly different from wild-type (*Figure 7E–F* and data not shown). Quantification of pseudopod length, using an aspect ratio measurement to normalize for variation in cell size (*Batchelder et al., 2011*), confirmed that Pronase-treated *comp-1* cells were significantly shorter than either wild-type or TEA-treated *comp-1* cells (*Figure 7B*). Since Pronase-treated *comp-1* cells contained distinct cell body and pseudopod regions, with normal localization of organelles (*Figure 5*, *Figure 7*, and data not shown), it is likely that these cells were polarized but failed to extend their pseudopods appropriately. Similar to other amoeboid cells, locomotion of

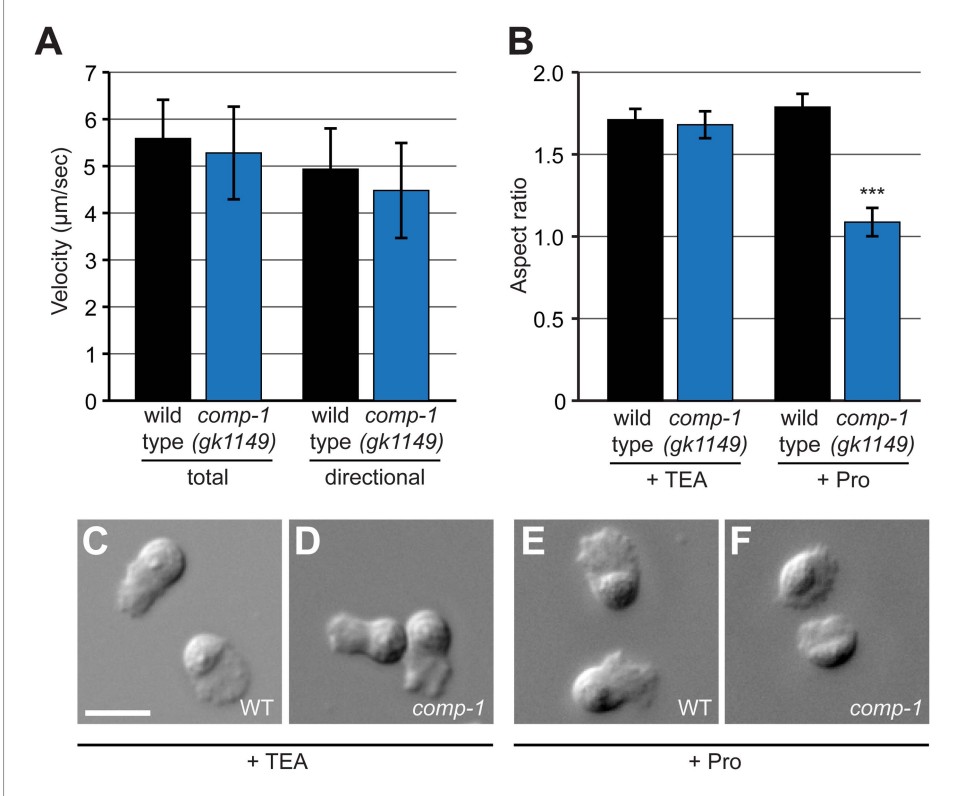

**Figure 7**. *comp-1* male sperm can migrate normally but have context-dependent defects in cell morphology. (**A**) *comp-1(gk1149)* sperm can migrate in vivo at speeds equivalent to wild-type sperm. Mitotracker-labeled males were crossed to N2 hermaphrodites and time-lapse images of sperm migrating through zone 2 were collected. Velocity and directional velocity toward the spermatheca were measured using ImageJ. (**B–F**) *comp-1(gk1149)* spermatids show reduced pseudopodial extension after activation by Pronase. (**B**) Quantification of aspect ratios of wild-type and *comp-1(gk1149)* sperm treated with either TEA or Pronase. (**C–F**) Representative images of wild-type (**C, E**) and *comp-1(gk1149)* (**D, F**) sperm treated with TEA (**C,D**) or Pronase (**E,F**). Error bars, 95% confidence interval; ***, $p < 0.001$, Kolmogorov–Smirnov test. Scale bar, 5 μm.

The following figure supplement is available for figure 7:

**Figure supplement 1**. *comp-1* sperm can crawl and be activated in vitro.

nematode sperm depends on protrusion of the lamellipodium-like pseudopod, adhesion to substrate, and retraction of the cell body (*Roberts and Stewart, 2000*; *Bottino et al., 2002*). Pseudopod extension defects would be expected to result in altered locomotion and/or interactions with the hermaphrodite reproductive tract, which in turn should affect migration to and occupation of the spermathecae.

## Discussion

Taking advantage of the male-hermaphrodite reproductive system of *C. elegans* and its robust natural male precedence order, we have used a genetic screen to identify a sperm competition mutant, *comp-1*. While mutant sperm are used at normal levels in non-competitive contexts, they display severe usage defects in all competitive contexts. When wild-type sperm are present, *comp-1* sperm are largely absent from the spermathecae and thus are virtually excluded from opportunities to fertilize oocytes. This usage pattern leads to severe defects in male reproductive success for males as well as failure to benefit from outcrossing for hermaphrodites. Consistent with their localization defects in vivo, *comp-1* sperm have in vitro defects in pseudopodial extension that, like their usage defects, are dependent on context. Together, these phenotypes suggest a cellular role for *comp-1* in

modulating the response of sperm to their environment. To our knowledge, *comp-1* is the first gene identified in *C. elegans* to specifically regulate sperm competition and one of few implicated in this process in any organism. Its pattern of conservation in related species suggests a role in male–male sperm competition outside the male-hermaphrodite mode of reproduction used by *C. elegans*. Our findings demonstrate that functional traits can influence the outcome of sperm competition in *C. elegans* in a manner independent of sperm size.

### *comp-1* and sperm success

For males, reproductive success depends on several aspects of sperm function. To fertilize oocytes, sperm must be transferred, become motile, and migrate to the site of fertilization in response to guidance signals (*Ward and Carrel, 1979*; *Kubagawa et al., 2006*). Overall fecundity depends on the number of sperm that accomplish these behaviors as well as their ability to be stored so as to ensure long-term usage (*Murray and Cutter, 2011*). We have found that *comp-1* sperm are transferred at rates comparable to the wild type, so they achieve initial entry into the reproductive tract, but they then show varying defects in the ensuing steps (*Figure 8*). Although *comp-1* sperm show delays in migration toward the spermathecae, at least some sperm migrate rapidly and directionally, arguing against a defect in locomotion per se. Large numbers eventually accumulate in the spermathecal region, suggesting that they respond to directional cues, but they are generally found outside the spermathecal valve. Once self-sperm stores are depleted, *comp-1* sperm concomitantly gain

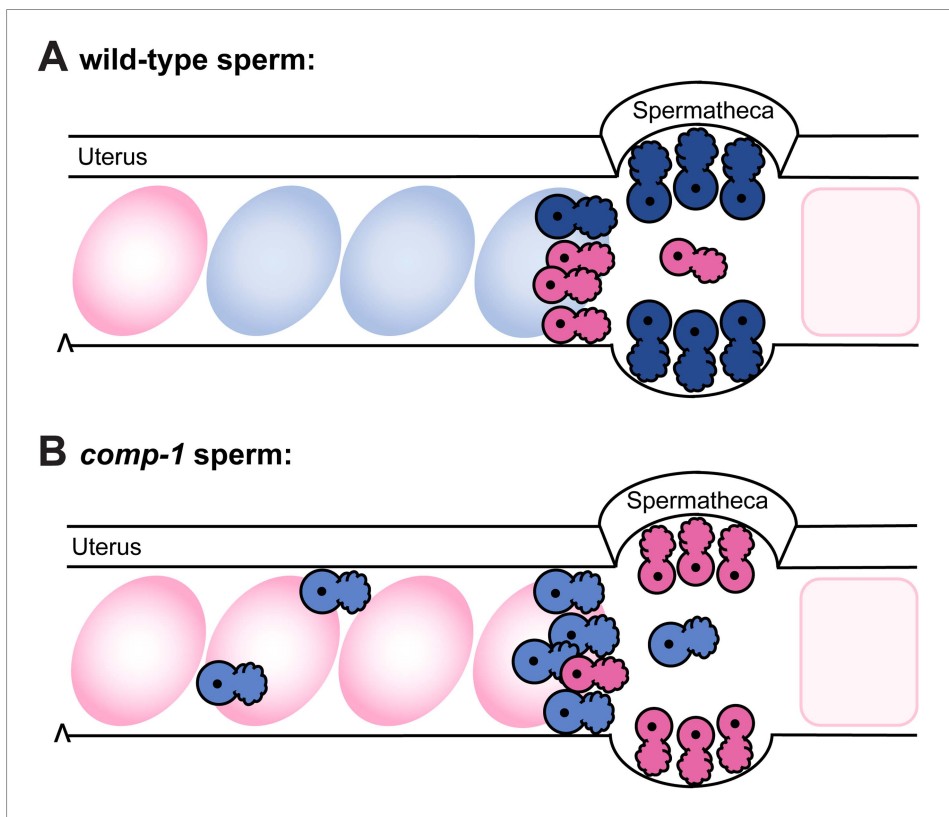

**Figure 8**. Model: *comp-1* sperm have localization defects that result in failure to compete with wild-type sperm. (**A**) Wild-type male sperm (blue) migrate to the region of the spermathecae, where they displace hermaphrodite self sperm (pink) and preferentially fertilize oocytes. Oocytes fertilized by male sperm are shown in blue; oocytes fertilized by self sperm are shown in pink. (**B**) *comp-1* mutant male sperm (light blue) migrate to the spermathecae, but remain outside while wild-type sperm (pink) are present, and are thus excluded from opportunities to fertilize oocytes. They also show delayed migration to the spermathecal region and increased localization in the periphery of the female reproductive tract.

residency in the spermathecae and begin to fertilize oocytes. Since fertilization occurs only in these structures, it is likely that this localization defect underlies the reduced competitive ability and generally poor reproductive success of *comp-1* males.

Spermathecal occupancy depends on the balance between the rate of entry due to migration and the rate of loss due to displacement by oocytes, which rearrange and even expel a subset of stored sperm as they pass through during ovulation (*Ward, 1977*; *Ward and Carrel, 1979*). Male sperm could thus increase their numbers in the spermathecae either by resisting removal, for example, by increasing adhesion to the spermathecal walls, and/or by migrating quickly back into the spermathecae, for example, by increasing their crawling velocity. Defects in pseudopodial extension like those observed in vitro for *comp-1* could affect either of these processes, allowing wild-type sperm to preferentially associate. Future studies will be necessary to differentiate between the two models, as well as to characterize the dynamics of sperm behavior in storage.

A unique feature of the *comp-1* phenotype is its dependence on the presence of wild-type sperm in recipient hermaphrodites. Do *comp-1* sperm defects occur because wild-type sperm behavior is superior, leading to their physical displacement, or does the presence of wild-type sperm make *comp-1* sperm inferior, through an indirect mechanism such as signaling? In some organisms, sperm may cooperate by associating with one another to promote fertility or by providing different functions within an ejaculate (reviewed in *Higginson and Pitnick, 2010*), but neither cooperative nor detrimental interactions between sperm have been described for *C. elegans*. Some defects in localization of *comp-1* sperm are observed in the absence of competing cells; a few sperm can be found scattered throughout the uterus, though this mislocalization apparently does not lead to significant reduction in usage or loss from the reproductive tract. These findings are consistent with the migration and localization defects we observed in competitive contexts, and they indicate that *comp-1* defects are not solely induced by the presence of wild-type sperm. However, we cannot exclude the possibility that sperm–sperm interactions influence the outcome of competition between *comp-1* and wild-type cells.

Several studies have demonstrated a strong association between precedence and cell size in *C. elegans* (*LaMunyon and Ward, 1998*, *1999*; *Murray et al., 2011*). However, loss of *comp-1* has no effect on cell size. Furthermore, *comp-1* activity appears to override the contribution of size, since large *comp-1* male sperm are completely outcompeted by small wild-type hermaphrodite sperm. Interestingly, the normal male precedence order is restored when both male and hermaphrodite sperm lack *comp-1* function, consistent with the idea that the size effect again predominates. Our data thus suggest that multiple activities contribute to precedence and can be independently modulated to affect sperm competitive ability. We note that a mechanism involving altering the activity of COMP-1 is likely to be less costly than production of larger sperm, which is associated with a reduced rate of sperm production (*LaMunyon and Ward, 1998*; *Murray et al., 2011*).

## The cellular role of COMP-1

How does COMP-1 function in sperm to alter motility-related behaviors? For crawling cells, locomotion and interaction with substrate are dependent on maintenance of polarity and extension of the lamellipodium or the pseudopod in the case of nematode sperm (reviewed in *Lammermann and Sixt, 2009*; *Reig et al., 2014*). *C. elegans* sperm are stably polarized, though the shape and size of their pseudopods is dynamically regulated (*Nelson et al., 1982*). Markers of the cell body and pseudopod are appropriately localized in *comp-1* sperm, suggesting polarity is not disrupted. However, treatment with Pronase in vitro, which is thought to mimic the endogenous male activator (*Smith and Stanfield, 2011*), generates activated cells with severely shortened pseudopods. The sperm cytoskeleton lacks actin and instead consists of Major Sperm Protein (MSP), which generates a network of fibers that drives cell protrusion via its expansion and contraction (*Italiano et al., 1999*; *Roberts and Stewart, 2012*). In the related nematode *Ascaris*, MSP filament assembly is mediated by MPOP, a pH-dependent phosphoprotein that is active at the leading edge (*LeClaire et al., 2003*) and the soluble proteins MFP1 and MFP2 (*Buttery et al., 2003*). MSP dynamics are also governed in part by the PP1 phosphatase GSP-3/4, which localizes to the proximal pseudopod near the cell body (*Wu et al., 2011*). Since COMP-1 localizes to the cell body, it seems unlikely to interact directly with the MSP cytoskeleton, but rather might function upstream of locomotion per se. COMP-1 contains a protein kinase-like domain, which might suggest a role in signal transduction. Like many other

reproductive proteins, it represents a divergent member of its family, and its primary sequence suggests that it is unlikely to be catalytically active. However, in spite of lacking or having reduced enzymatic activity, pseudokinases have been shown to play important roles in cell signaling via interactions with active kinases or their substrates, scaffolding or tethering of signaling complexes, and other mechanisms (reviewed in *Reiterer et al., 2014*). The punctate localization of COMP-1 within the sperm cell body is intriguing in this context.

Our finding that *comp-1* sperm have reduced pseudopod lengths in an in vitro assay fits with their altered patterns of localization in vivo. However, measurements of cell velocity indicate that cells lacking *comp-1* are capable of wild-type crawling speeds and they eventually accumulate near their appropriate target. Therefore, it is probable that the cellular defects of *comp-1* sperm in vivo are less severe than those of Pronase-treated *comp-1* sperm, which have severely shortened pseudopods and should be nearly incapable of movement (*Nelson et al., 1982*; *LaMunyon and Ward, 1999*). The *comp-1* phenotype is also distinct from that caused by lack of prostaglandin cues involved in guidance toward oocytes, which leads to a severe reduction in crawling velocity along with loss of directionality (*Kubagawa et al., 2006*; *Edmonds et al., 2010*). Thus, *comp-1* sperm are capable of directional migration, though some aspect of sensing or responding to prostaglandins could be impaired. Alternatively, the altered localization of *comp-1* sperm could stem from decreased adhesion to the substrate, leading to a reduced ability to crawl directionally and/or maintain position within the spermathecae. Overall, the context dependence of *comp-1* sperm usage suggests that cellular defects may be limited to a subset of sperm cells or may be manifested only some of the time, for example during interaction with particular substrates within the reproductive tract. Sperm migrate across a variety of tissues including uterine and spermathecal cells and fertilized eggs, each of which could be more or less permissive for migration of *comp-1* mutant cells due to effects on either adhesion or signaling.

## *comp-1* and reproductive success

The role of *comp-1* in *C. elegans* is evident by the reduction in reproductive success for both sexes in crosses to *comp-1* males. Wild-type males who mate successfully can produce hundreds (up to thousands) of offspring (*Wegewitz et al., 2008*), but *comp-1* males produce very few cross progeny, and these are delayed until other sperm are no longer available. Hermaphrodites mated to wild-type males increase their overall progeny production, but this increase is significantly lower in crosses to *comp-1* males, and few cross progeny are generated. Even in crosses between *comp-1* males and *comp-1* hermaphrodites, where the male precedence order is largely restored, males show reduced success as compared with wild-type to wild-type matings. Thus, sperm with *comp-1* function should be highly selected for usage when competing with sperm without *comp-1*. In male-female species, we expect that *comp-1* may have a similar function in improving male reproductive success, depending on the rate of polyandry in a given population.

Although self fertilization allows *C. elegans* to propagate without the need to mate and eliminates the cost of producing males, it also leads to reduced genetic variation (discussed in *Anderson et al., 2010*). The rate of outcrossing in wild populations is estimated to be low, yet males exist, suggesting that some outcrossing may be selected for, or alternatively, that androdioecy has arisen sufficiently recently that the specialized developmental and behavioral characteristics of males have not had time to degrade. Selective pressure has been shown to increase the rate of outcrossing in *C. elegans* in several experimental schemes (*Lopes et al., 2008*; *Morran et al., 2009a*; *Morran et al., 2009b*; *Anderson et al., 2010*). By promoting the preferential usage of male sperm, COMP-1 should function to increase the genetic diversity of offspring and thus may confer a fitness benefit in situations where adaptation is beneficial (*Carvalho et al., 2014*).

## Sperm competition in *C. elegans*

The outcome of sperm competition depends on the arena in which it occurs, which depends on the specialized reproductive biology and anatomy of the species in question. In particular, differences in the capacity of the sperm storage organ(s), functional characteristics of sperm and seminal fluid, and the degree of sperm mixing lead to distinct patterns of sperm usage (*Parker and Pizzari, 2010*). In *C. elegans*, the spermathecae are somewhat limited as storage sites, which likely reduces the incentive for males to produce and transfer vast numbers of sperm. Instead, the arms race between

the sexes leads to males producing sperm that are functionally superior. Once male sperm reach the spermathecae, they are immediately used even though they lack numerical superiority (GMS, unpublished data). Interactions between competing ejaculates can be divided into offense, the ability to displace previous sperm, and defense, the ability to block subsequent sperm. Observations of the processes of ovulation, sperm migration, and fertilization in wild-type *C. elegans*, as well as the ability of fertilization-incompetent sperm to sterilize hermaphrodites, suggest that wild-type male sperm most likely block the access of self sperm to the site of fertilization (*Ward and Carrel, 1979*; *Singson et al., 1999*). However, they fail to block the sperm of another male, as no precedence order is observed in sequential wild-type matings (*Ward and Carrel, 1979*; *LaMunyon and Ward, 1998*). *comp-1* male sperm lack the ability to suppress self progeny production, and they also show severe defects in male–male competition whether they are the first or a subsequent mate. Thus, they appear to totally lack the offensive capabilities of normal sperm and also show defects in defense against new rivals.

COMP-1 is present in both male-hermaphrodite and male-female species of nematodes. Since the male–female reproductive mode is ancestral (*Kiontke et al., 2004*; *Cutter et al., 2008*), the function of COMP-1 in sperm competition most likely originated in male–male competition and has been retained in androdioecious species, such as *C. elegans*, where it remains necessary for both male–male and male-hermaphrodite sperm competition. Our results thus establish that *C. elegans* provides a general model to study the molecular mechanisms that underlie sperm competition as well as the interplay between the cell biology of sperm and the forces of sexual selection.

## Materials and methods

### *C. elegans* culture and strains

*C. elegans* strains were grown at 20°C, except where noted, and fed with OP50 *Escherichia coli* bacteria as previously described (*Brenner, 1974*). All strains were derived from the N2 Bristol wild-type strain, with the exception of the CB4856 Hawaiian strain used for mapping. For experiments involving males, *him-5* strains were used as our wild-type: *him-5(e1490)* was used for the genetic screen and *him-5(ok1896)* was present in all other strains from which males were obtained (*Hodgkin et al., 1979*), unless explicitly noted. *comp-1(me69)* was identified in this study and the *comp-1* (*gk1149*) allele was generated by the C. elegans *Deletion Mutant Consortium (2012)*. Other alleles used for experiments were *spe-8(hc40,hc53)* I, *mIs11[myo-2::GFP, pes-10::GFP and gut::GFP]*, *ttTi5605* II, *oxSi221[Peft-3::GFP]* II, *unc-119(ed3)* III, *fem-3(q20gf)* IV, *dpy-4(e1166)* IV, *cxTi10816* IV, *fog-2(q71)* V, and *him-5(e1490, ok1896)* V (*Wood and the Community of* C. elegans *Researchers, 1988*; *Maduro and Pilgrim, 1995*; *Frøkjær-Jensen et al., 2008*, *2012*; *Meneely et al., 2012*).

To generate transgenic strains, Mos-mediated Single Copy Insertion (MosSCI) was used to integrate transgenes at the *ttTi5605* II and *cxTi10816* IV loci (*Frøkjær-Jensen et al., 2008*, *2012*).

### Genetic screen and identification of *comp-1*

The *me69* mutant was isolated in a screen for males with reduced sperm precedence or fertility. *him-5 (e1490)* hermaphrodites were mutagenized using ethyl methanesulfonate (EMS) mutagenesis as described in *Wood and The Community of* C. elegans *Researchers, 1988*. Groups of 7–8 P0 hermaphrodites were allowed to self-fertilize; L4 F1 hermaphrodites were picked (25 per plate); and individual L4 F2s were used to establish lines potentially homozygous for newly induced mutations. To assay male precedence, from each viable line 4–5 L4 males were mated to one *spe-8(hc40); dpy-4* hermaphrodite for approximately 48 hr, at which time the cross was terminated by removing the hermaphrodite. When all progeny reached at least the L4 stage, mating plates were examined. If at least 5 Dumpy (self) progeny were present, the number of Dumpy (self) and non-Dumpy (cross) progeny were counted. Such lines were retested using the same precedence assay as before. Approximately 3400 mutagenized lines were tested and 16 lines were recovered as homozygous mutants. Of the 16 lines, six lines had normal gonadal and sperm morphology, consistent with a precedence-specific defect. The *me69* mutant was among those 6 lines.

To map *me69*, CB4856 (Hawaiian) males were crossed to *me69; him-5* hermaphrodites, F1 males were crossed back to *me69; him-5* hermaphrodites, and individual F2 males were tested for the male precedence defect. Each male was recovered into lysis buffer and males scoring as mutant were assayed for a centrally located SNP on each chromosome (*Wicks et al., 2001*). Linkage was detected

to chromosome I and additional SNPs were scored in individual males to narrow *me69* to a 6.7-Mb region between WBVar00240399 and WBVar00240414 (*Tables 1 and 2*) (*WormBase*; *Jakubowski and Kornfeld, 1999*). To identify the gene affected in *me69*, whole genome sequence was obtained from the strain isolated in our genetic screen. Of 45 variations in the *me69* region, 24 were consistent with EMS, seven affected coding regions, and only one affected a gene (*F37E3.3*) showing sperm-enriched gene expression.

## Molecular biology

Molecular biology was performed according to standard protocols. The Multisite Gateway Three-Fragment Vector Construction Kit (Life Technologies, Grand Island, NY) was used to construct donor plasmids. Fragments were then recombined into the MosSCI destination vectors pCFJ150 or pCFJ212 (*Frøkjær-Jensen et al., 2008*, *2012*). For constructs in which two fragments were ligated by PCR, fusion PCR was performed as in *Hobert (2002)*. Primers used for generating constructs are listed in *Table 3*, and plasmid construction strategies are summarized in *Table 4*.

## Fertility and sperm competition assays

To measure hermaphrodite fertility, L4 hermaphrodites were individually placed on a freshly seeded lawn and moved to a new plate every 24 hr until eggs were no longer laid. To measure male fertility, L4 males were crossed in a 1:1 ratio to L4 *fog-2* females for 24 hr. The males were then removed and the females were transferred every 24 hr until egg laying ceased. Progeny were counted after reaching at least the L4 stage. The variability in cross progeny number observed in these experiments is typical of this assay and is generally attributed to variation in mating, sperm transfer, and/or sperm loss (*Murray et al., 2011*).

To test short-term male precedence, L4 males and *spe-8(hc53); dpy-4* or *dpy-4* L4 hermaphrodites were placed together in a 1:1 ratio onto plates with freshly seeded lawns. After 40 hr, both parents were removed. Upon reaching adulthood, offspring were scored as either Dumpy (self) or non-Dumpy (cross) progeny and counted. To test long-term male precedence, animals were allowed to mate for 16 hr, hermaphrodites were transferred to fresh plates every 12 hr, and self and cross progeny were scored as described above. To test the effect of hermaphrodite age on male precedence, 12, 24, 36, or 48 hr post-L4 hermaphrodites were crossed to 24 hr post-L4 males for 24 hr, both parents were removed, and the number of self and cross progeny were scored as described above. To estimate the number of self sperm remaining in the hermaphrodite reproductive tract at each time point, the number of progeny from unmated hermaphrodites picked in parallel was counted.

Male–male competition assays were performed by placing 24 hr post-L4 adult males ('first' males) with 24 hr post-L4 adult *fog-2(q71)* hermaphrodites for 3 hr in an 8:6 ratio of males to hermaphrodites. The hermaphrodites were allowed to recover for 1 hr, and those lacking visible embryos in their uteri

**Table 1**. *me69* is linked to chromosome I

| Marker* | Genetic position* | Genomic position† | Haw/+ frequency‡ |
|---|---|---|---|
| WBVar00240399 | I:0.91 | I:6350803 | 1/16 |
| WBVar00172772 | II:0.12 | II:6789208 | 8/16 |
| WBVar00067953 | III:−0.31 | III:8318640 | 10/16 |
| WBVar00188750 | IV:1 | IV:4625317 | 3/16 |
| WBVar00240687 | V:0.88 | V:8177520 | 9/16 |

*Wicks et al. (2001)*.
†WormBase WS243 (accessed 30 August 2014).
‡*me69; him-5* males were crossed to CB4856 Hawaiian hermaphrodites, F1 males were crossed back to *me69; him-5* hermaphrodites, and F2 males were assayed for precedence defects in crosses to *spe-8; dpy-4* hermaphrodites. Animals scoring as mutant (*me69* homozygotes) were scored by PCR and restriction digest for centrally-located SNPs on each chromosome. Animals lacking Hawaiian alleles at all loci tested were considered self progeny and excluded from analysis.

**Table 2**. Mapping of *me69* on chromosome I

| No. F2s* | WBVar 00240 394† 825026 | WBVar 00240 397 5482531 | WBVar 00240 399 6351803 | WBVar 00155 231 8646304 | WBVar 00240 416 10614690 | WBVar 00240 407 11472093 | WBVar 00159 097 12433167 | WBVar 00240 414 13066381 | WBVar 00161 629 14154889 |
|---|---|---|---|---|---|---|---|---|---|
| 16 | B/B | | | | B/B | | | | B/B |
| 6 | H/B | | | | B/B | | | | B/B |
| 2 | H/B | B/B | B/B | B/B | B/B | B/B | B/B | H/B | H/B |
| 3 | B/B | B/B | B/B | B/B | B/B | B/B | B/B | H/B | H/B |
| 1 | H/B | H/B | H/B | B/B | B/B | B/B | B/B | B/B | B/B |

F2 males from the cross described in **Table 1** were scored for SNPs across chromosome I. Animals were either homozygous Bristol (B/B) or heterozygous for the Hawaiian allele (H/B) at each SNP.

*Number of F2 males showing each pattern.

†SNP designation and genomic position on chromosome I. **Wicks et al. (2001)**; **WormBase**.

were removed from the plate. The 'second' males, 28 hr post-L4, were then placed with the hermaphrodites and allowed to mate for 3 hr. Individual hermaphrodites were then moved to fresh plates, allowed to lay eggs for 16 hr, then transferred. To distinguish progeny of first and second mates from one another, second-male strains harbored an integrated GFP transgene, *mIs11*; to control for possible marker-specific effects, experiments were repeated with the *mIs11* strains as first males. Progeny generated 0–16 hr after mating were quantified. Subsequent progeny were scored for GFP, and only plates that contained both GFP-positive and GFP-negative offspring were included in analyses.

Male mating efficiency was assessed based on two of the assays described above. The frequency of observing successful sperm transfer and/or offspring production in crosses to *spe-8; dpy-4* hermaphrodites was 74–88% for wild type and 48–61% for *comp-1*. The frequency of offspring production in crosses to *fog-2* females was 85–100% for wild type, 60–85% for *comp-1(me69)*, and 55–90% for *comp-1(gk1149)*. For all experiments involving measurements of progeny numbers, wild-type and mutant animals were tested in parallel to control for variations in temperature and/or media

**Table 3**. Construction of entry plasmids used to generate targeting constructs

| Fragment description | Fragment length | Forward primer | Reverse primer | Vector | Plasmid name |
|---|---|---|---|---|---|
| *comp-1* promoter | 712 | GGGACAACTTTGTATA GAAAAGTTGCCAGTTC CTCGCCTAGCTTTC | GGGACTGCTTTTTTGTAC AAACTTGATGCTTTTGAT TCGATAGATGATCC | pDONR P4-P1r | pJMH1 |
| *comp-1* coding region | 1921 | GGGGACAAGTTTGTACAAA AAAGCAGGCTCAATGACG TTGGTCGAATCGAAAC | GGGACCACTTTGTACA AGAAAGCTGGGTCTTA TTTGCGCTGGAATTGATC | pDONR 221 | pJMH2 |
| *comp-1* coding region without stop codon | 1918 | GGGGACAAGTTTGTACAAAA AAGCAGGCTCAATGACGTT GTCGAATCGAAAC | GGGACCACTTTGTACAAG AAAGCTGGGTATTTGCG CTGGAATTGATC | pDONR 221 | pJMH3 |
| *comp-1* 3′ region | 561 | GGGGACAGCTTTCTTGTAC AAAGTGGAAGAACTTA CGGAAGAATATG | GGGGACAACTTTGTATAAT AAAGTTGATGCGTTCTC ATCAGGCTTC | pDONR P2r-P3 | pJMH4 |
| *peel-1* coding region* without stop codon | 3279 | GGGGACAAGTTTG TACAAAAAAGCAGGCTG CTTAATGCGCTTTGGTAAG | GGGGACCACTTTGTACA AGAAAGCTGGGTCTGGATT TTCAACACTTGGATC | pDONR 221 | pJMH20 |

***Seidel et al. (2011)**.

**Table 4**. Description of targeting constructs used to generate transgenic worm strains

| Construct | Position 1 pDONR P4-P1r | Position 2 pDONR 221 | Position 3 pDONR P2r-P3 | Destination vector | Locus |
|---|---|---|---|---|---|
| Pcomp-1::comp-1::comp-1 3' region | pJMH1 | pJMH2 | pJMH4 | pCFJ150* | ttTi5605 |
| Pcomp-1::H2B::GFP::comp-1 3' region | pJMH1 | pCM1.35† | pJMH4 | pCFJ150 | ttTi5605 |
| Ppeel-1::comp-1::tbb-2 3' region | Ppeel-1 [4-1]‡ | pJMH2 | pCM1.36† | pCFJ150 | ttTi5605 |
| Ppeel-1::comp-1::GFP::unc-54 3' region | Ppeel-1 [4-1] | pJMH3 | pGH50§ | pCFJ150 | ttTi5605 |
| Ppeel-1::comp-1::mCherry::unc-54 3' region | Ppeel-1 [4-1] | pJMH3 | mCherry::unc-54 3' region§ | pCFJ150 | ttTi5605 |
| Ppeel-1::peel-1::GFP::unc-54 3' region | Ppeel-1 [4-1] | pJMH20 | pGH50 | pCFJ212* | cxTi10816 |

*Frøkjær-Jensen et al. (2008).
†Merritt et al. (2008).
‡Seidel et al. (2011).
§Liu et al. (2009).

quality that can affect mating and fertility. Each experiment was repeated 2–4 times, and figures show representative results.

## Microscopy and immunohistochemistry

To release spermatids, adults were dissected in a drop of sperm medium (SM; 50 mM HEPES pH7.8, 50 mM NaCl, 25 mM KCl, 1 mM MgSO$_4$, 5 mM CaCl$_2$, and 10 mM dextrose). Virgin 48 hr post-L4 males grown at 20°C were used. Where necessary, spermatids were incubated in SM containing 60 mM TEA or 200 µg/ml Pronase to induce activation into motile sperm (*Shakes and Ward, 1989*). Antibody staining followed a protocol similar to that in *Wu et al. (2011)*. Briefly, an equal volume of 4% paraformaldehyde in SM was added to the dissected cells. The slides were then incubated in a humid chamber for 5 min, freeze-cracked on a metal block placed in liquid nitrogen, incubated in 95% ethanol for 1 min, and washed with PBST (phosphate-buffered saline pH 7.2, 0.5% Triton X-100, 1 mM EDTA). Antibody incubations were performed for 16 hr at 4°C with rabbit anti-GSP-3/4 (rb1496, 1:500) (*Wu et al., 2011*) and 1 µg/mL DAPI (4',6-diamidino-2-phenylindole) and for 2 hr at 4°C in goat anti-rabbit AlexaFluor 488- or AlexaFluor 568-labeled IgG (Life Technologies) at 1:1000; antibodies were diluted and washes were performed in PBST with 1% BSA (bovine serum albumin). Slides were mounted with VectaShield (Vector Laboratories, Burlingame, CA). Confocal images were acquired using an Olympus FV1000 confocal microscope.

## in vivo sperm migration and localization assays

To analyze localization of male sperm up to 2.5 hr after transfer, Mitotracker Red CMXRos (Life Technologies) was used to label male sperm as in *Stanfield and Villeneuve, 2006*. To analyze sperm localization more than 2.5 hr after transfer, virgin 24 hr post-L4 males carrying the *Pcomp-1::GFP::H2B* transcriptional reporter were mated for 45 min to 24 hr post-L4 N2 hermaphrodites anesthetized in 0.1% tricaine and 0.01% tetramisole (*McCarter et al., 1997*), and males were then removed. At 12 hr or 24 hr post-mating, images of each recipient were captured in multiple focal planes to capture an entire gonad arm. Analysis of sperm position was performed as in *Edmonds et al. (2010)*. Depending on the experiment, either all GFP-positive male sperm in a gonad arm were counted, or those in the focal plane that had the most sperm in the spermatheca were counted. To analyze localization of self sperm, 24 hr post-L4 hermaphrodites were fixed with Carnoy's fixative (*Ellis and Horvitz, 1986*) and stained with DAPI at 1 µg/ml in M9. Image collection and data analysis were performed as for male sperm.

To measure in vivo velocity, images of migrating cells were collected as in *Kubagawa et al. (2006)* using an AxioImager M1 microscope, Axiocam camera, and Axiovision software (Zeiss, Germany). Cells within Zone 2 that moved for at least four consecutive frames were analyzed using the plugins Manual Tracking and Chemotaxis and Migration Tool (Ibidi, Germany) in ImageJ (*Schneider et al., 2012*).

## in vitro sperm morphology and function assays

Sperm were activated in vitro as described previously and DIC (differential interference contrast) images were captured every 60 s for 30 min (*Shakes and Ward, 1989*; *Fenker et al., 2014*). To quantify activation, sperm were scored for the presence of either spikes or a pseudopod at 30 min after adding Pronase. Non-activated sperm from control slides lacking activator were used to measure spermatid size. Aspect ratio was measured by dividing the total length of the pseudopod and cell body by the width of the cell body. The center of the cell body was determined by fitting a circle or ellipse around the cell body and finding the center of that object. The length was then determined by drawing a line from the tip of the pseudopod to an edge of the cell body, with the line dissecting the center of the cell body, and the width of the cell was measured by drawing a line perpendicular to the length and dissecting the center of the cell body. Velocity was measured in TEA-activated sperm that moved for at least 3 consecutive frames. Measurements were obtained using ImageJ (*Schneider et al., 2012*).

## Acknowledgements

We thank K Fenker, T Shimko, and C Thummel for comments on the paper. We thank B Duffy for help with experiments and R Clark and C Thacker for assistance with sequencing. We thank D Chu, J Yanowitz, and numerous members of the Jorgensen lab for sharing reagents. Confocal microscopy was performed at the University of Utah Imaging Core. Some strains were provided by the CGC, which is funded by the NIH Office of Research Infrastructure Programs (P40 OD010440).

## Additional information

### Funding

| Funder | Grant reference | Author |
| --- | --- | --- |
| National Institutes of Health (NIH) | R01-GM087705 | Gillian M Stanfield |
| National Institutes of Health (NIH) | T32-GM007464 | Jody M Hansen |

The funders had no role in study design, data collection and interpretation, or the decision to submit the work for publication.

### Author contributions

JMH, DRC, GMS, Conception and design, Acquisition of data, Analysis and interpretation of data, Drafting or revising the article

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
