## [Decision Letter]

Thank you for sending your work entitled “COMP-1 promotes competitive advantage of nematode sperm” for consideration at *eLife*. Your article has been favorably evaluated by Diethard Tautz (Senior editor), Oliver Hobert (Reviewing editor), and two reviewers.

All of the reviewers find that this manuscript reports very interesting results and agree that the results are very well presented. Both reviewers only have a number of minor comments, which you will find listed below. Based on the reviewers’ very positive comments, we will be happy to consider a revised version of the manuscript in which we ask you to please address the minor comments by the reviewers. It will not be necessary to provide a detailed response to the reviewers’ comments and we trust that you will incorporate these minor points.

1) In the Introduction, the phrase “defensive and offensive” should be defined explicitly. Some readers might not realize it refers to sperm from the first male defending their priority against offensive sperm from the second male. In the context of worm biology, it is not clear that these two ideas can be disentangled, since all sperm compete to reestablish positions in the spermatheca after the passage of each oocyte.

2) Figure 1, etc. Although the Kolmogorov-Smirnov test can be used in this situation, the Mann-Whitney U test provides a better way for comparing distributions with such different medians. Our rough calculations indicate that the P < 0.0001 in Figure 1, which is much better than the author's indicate. We recommend switching to this statistical test for Figure 1, and in similar examples later in the paper. Otherwise, the authors are significantly underestimating the significance of their results. There is a simple Mann-Whitney calculator at the Vassar web site.

3) Figure 1—figure supplement 1. It seems likely that there was a sequencing error in the *C.* sp. 5 assembly, resulting in a poor prediction for the COMP-1 protein structure. In the following genome sequence, the red letters are likely to be a small insertion caused by an assembly error: tttttgcagaaataacaaagtggaacttggaaatacattggagaagtttctcaaaaaccgggaggagcaattggatttgactcagagaatcaagttctgttcttctgccgtccgtattctctcggaacttcatcaatcggatatctatcatggagcatcgcaaatggaaaacttctatgttgagtttgctgggctcaaaccgaagacaatgaagaattacgagttgattttcaatggagcaaacggattgcttgttcaggggtagaacagtgttgaagggcattcaaactcaaatttcgatttcagaaaatccgataacactgttcgtgtcgtcgattacgattcgactgctccagaagttgcattcactcggaaattgtcggaaattgacgaaggagagtggggttttcaatttgggacgtcttttcgaacagattctgagaccagatctcatcaaatcgtacaaggattgccagggtaattcaagagttctttaattcagattctctagtcgatctcttttcagaagaacctcgttctctcaacgaaatgcgtcatctggttgctcgtgcaactcatccgaatcccactcgtcgtccaacaatgcatggcgtcgtcattatggttcgagatatt. If the red letters were deleted, the *C.* sp. 5 COMP-1 structure would be very similar to that of *C. remanei*. The authors should amplify and sequence this fragment to determine if *C.* sp. 5 has undergone a surprising evolutionary divergence, or if there is an assembly error in the current genome sequence.

4) Figure 1—figure supplement 1. The sequence for the N-terminus of *N. americanus* COMP-1 appears to be present in the genome sequence at Wormbase, and should be used to make a complete protein prediction.

5) At the end of the third paragraph of the Results section, please define how you calculated the range of percentages here.

6) In the subsection “*comp-1* male sperm are not used until hermaphrodite self sperm are depleted” change the second sentence to read: “we wanted to know if *comp-1* sperm were lost or remained active within the gonads of aging hermaphrodites”.

7) Figure 7: if sperm is dissected out from *comp-1* hermaphrodites, do they look like normal sperm or the pronase-activated sperm in panel F?

8) The Introduction is scholarly and complete, but I am wondering if it could be a little shorter.

9) Part of the decrease in *comp-1* mutant male competitiveness could be due to reduced ability to transfer sperm 48-85% (mutant) vs 74-100% (wild-type). Assuming no easily detectable somatic defects, have male mating behaviors (response, scanning, turning, spicule insertion) been closely examined?

10) *comp-1* sperm seem competent to respond to directional cues. Likely from PUFA derived prostaglandin-like signals. To nail this down, would it be useful to look at migration in *fat* mutants? Maybe the pull is there but not quite as strong. Alternatively, *comp-1* mutant sperm have a harder time entering the spermatheca. I wonder if *comp-1* mutants could hint at a capacitation-like process in worms. Sperm would need to have physiological changes as they progress closer to the site of fertilization.

---

## [Author Response]

*1) In the Introduction, the phrase “defensive and offensive” should be defined explicitly. Some readers might not realize it refers to sperm from the first male defending their priority against offensive sperm from the second male. In the context of worm biology, it is not clear that these two ideas can be disentangled, since all sperm compete to reestablish positions in the spermatheca after the passage of each oocyte*.

As these classifications for competitive ability are not commonly applied to sperm competition in *C. elegans*, and may be informative primarily to specialists in the field, we deleted this terminology from the Introduction and Results. However, we feel it is still appropriate to include these concepts in the Discussion, so we have ensured that the terms are defined in that section of the paper.

*2)*
Figure 1*, etc. Although the Kolmogorov-Smirnov test can be used in this situation, the Mann-Whitney U test provides a better way for comparing distributions with such different medians. Our rough calculations indicate that the P < 0.0001 in*
Figure 1*, which is much better than the author's indicate. We recommend switching to this statistical test for*
Figure 1*, and in similar examples later in the paper. Otherwise, the authors are significantly underestimating the significance of their results. There is a simple Mann-Whitney calculator at the Vassar web site*.

After careful consideration and review of our data, we have chosen to retain our original methods for statistical analyses. We agree that either the K-S or M-W test could be used. However, our understanding is that whereas the M-W primarily detects differences in the median and is better used to compare distributions with similar shape, the K-S test is sensitive to differences in either median or shape of a given distribution. Since the K-S test requires fewer assumptions about the underlying distributions involved, we believe it is most appropriate for the majority of our data. Systematic comparison of the two tests for our data sets found that they give completely concordant results in nearly all cases; where differences in p value cutoffs were observed, the K-S actually tended to give a lower p value. We hope that this explanation is sufficient to address questions about our use of the K-S test.

The reviewers’ rough calculation for Figure 1 is correct that the p value for Figure 1 should have been lower than that shown in the original submission; however, in fact, their analysis uncovered an error in which we had reported the significance values for a different technical repeat of that experiment other than the one shown in the figure. We thank the reviewers for detecting this error. The resubmitted version of Figure 1 contains the corrected p value, and values in other figures have been rechecked to confirm that they are accurate.

*3)*
Figure 1—figure supplement 1*. It seems likely that there was a sequencing error in the* C. *sp. 5 assembly, resulting in a poor prediction for the COMP-1 protein structure. In the following genome sequence, the red letters are likely to be a small insertion caused by an assembly error: tttttgcagaaataacaaagtggaacttggaaatacattggagaagtttctcaaaaaccgggaggagcaattggatttgactcagagaatcaagttctgttcttctgccgtccgtattctctcggaacttcatcaatcggatatctatcatggagcatcgcaaatggaaaacttctatgttgagtttgctgggctcaaaccgaagacaatgaagaattacgagttgattttcaatggagcaaacggattgcttgttcaggggtagaacagtgttgaagggcattcaaactcaaatttcgatttcagaaaatccgataacactgttcgtgtcgtcgattacgattcgactgctccagaagttgcattcactcggaaattgtcggaaattgacgaaggagagtggggttttcaatttgggacgtcttttcgaacagattctgagaccagatctcatcaaatcgtacaaggattgccagggtaattcaagagttctttaattcagattctctagtcgatctcttttcagaagaacctcgttctctcaacgaaatgcgtcatctggttgctcgtgcaactcatccgaatcccactcgtcgtccaacaatgcatggcgtcgtcattatggttcgagatatt. If the red letters were deleted, the* C. *sp. 5 COMP-1 structure would be very similar to that of* C. remanei*. The authors should amplify and sequence this fragment to determine if* C. *sp. 5 has undergone a surprising evolutionary divergence, or if there is an assembly error in the current genome sequence*.

We obtained sequence from the *Csi*-*comp-1* region, which indeed revealed a mis-assembled region. Using the corrected data, we generated a gene model with strong conservation across the length of COMP-1. The new protein alignment in Figure 1—figure supplement 1 incorporates this sequence.

*4)*
Figure 1—figure supplement 1*. The sequence for the N-terminus of* N. americanus *COMP-1 appears to be present in the genome sequence at Wormbase, and should be used to make a complete protein prediction*.

We manually inspected the *N. americanus* genome assembly and generated a protein prediction with strong conservation to *A. ceylanicum*, its nearest neighbor in our alignment, as well as similarity across the length of all COMP-1 orthologues. The new protein alignment in Figure 1—figure supplement 1 incorporates this sequence.

*5) At the end of the third paragraph of the Results section, please define how you calculated the range of percentages here*.

We used frequencies of cross-progeny production in 1:1 matings to *spe-8; dpy-4* hermaphrodites and to *fog-2* hermaphrodites, both situations in which progeny are only generated when mating and sperm and/or seminal fluid transfer occurs. This information has been added to the Methods.

Of note, other independent measurements of mating frequency were consistent with *comp-1* males having high mating frequencies that are similar to those of wild-type animals. For example, in experiments using mitotracker to visualize transferred sperm, we also observed that mating frequencies were similar for wild-type and *comp-1* males.

*6*) *In the subsection “*comp-1 *male sperm are not used until hermaphrodite self sperm are depleted” change the second sentence to read: “we wanted to know if* comp-1 *sperm were lost or remained active within the gonads of aging hermaphrodites”*.

We have altered the sentence as suggested, with the exception of the word “aging” which might place undue emphasis on the age of the recipients rather than the functional status of *comp-1* sperm.

*7)*
Figure 7*: if sperm is dissected out from* comp-1 *hermaphrodites, do they look like normal sperm or the pronase-activated sperm in panel F*?

We find that dissected (in vivo-activated) *comp-1* hermaphrodite sperm are polarized, have pseudopods, and appear similar to wild-type hermaphrodite sperm, though we have not quantified aspect ratios. Since hermaphrodite sperm are very small and have correspondingly shorter pseudopods than male sperm, male sperm should provide a more sensitive assay for alterations in cell morphology. In addition, we have shown that *comp-1* hermaphrodite sperm are functional and localize properly the majority of the time in non-competitive contexts, consistent with having functional pseudopods. Our data suggest that *comp-1*’s pseudopod extension defects, like other phenotypes, are context-dependent rather than sex-specific.

*8) The Introduction is scholarly and complete, but I am wondering if it could be a little shorter*.

Our goal was to succinctly present the broader context of our work along with key background from studying sperm competition in *C. elegans*. Our hope is that providing this framework will increase the accessibility of this study to those outside the field, so we would prefer not to shorten the Introduction significantly.

*9) Part of the decrease in* comp-1 *mutant male competitiveness could be due to reduced ability to transfer sperm 48-85% (mutant) vs 74-100% (wild-type). Assuming no easily detectable somatic defects, have male mating behaviors (response, scanning, turning, spicule insertion) been closely examined*?

Our *comp-1* strains do appear to have a slightly reduced overall mating efficiency, typically affecting 1 to 2 crosses from each experimental repeat, which usually comprises 20 or more total crosses. We have not quantified mating behaviors in our strains, for several reasons. 1) In experiments using mitotracker to visualize sperm, the quantity of transferred sperm, as assessed by fluorescence intensity, appears similar for wild-type and *comp-1* males, suggesting that 2) most of our assays allow us to exclude cases in which mating fails altogether: these include precedence and fertility assays using *spe-8, dpy-4* or *fog-2* recipients, male-male precedence assays, fertility time-course assays, migration assays, and counts of sperm in the spermatheca. 3) In the case of one of our assays, male-hermaphrodite precedence tests using *dpy-4* recipients, differences in mating efficiency would be expected to contribute to the magnitude of the phenotype we observe; but our conclusions do not rely on this assay alone. In summary, we believe that any effect due to mating is minor compared to the severe deficits in progeny production by *comp-1* males, which we observe even in cases where sperm transfer is directly confirmed.

*10)* comp-1 *sperm seem competent to respond to directional cues. Likely from PUFA derived prostaglandin-like signals. To nail this down, would it be useful to look at migration in* fat *mutants? Maybe the pull is there but not quite as strong. Alternatively,* comp-1 *mutant sperm have a harder time entering the spermatheca. I wonder if* comp-1 *mutants could hint at a capacitation-like process in worms. Sperm would need to have physiological changes as they progress closer to the site of fertilization*.

This is an interesting question, and experiments to determine the contribution of the hermaphrodite environment to sperm competition are ongoing in our lab. Since including this line of research would require a major expansion of the work reported in this manuscript, we have not incorporated those data in our resubmission.